# Drainage divide networks, Part 1: Identification and ordering in digital elevation models

Dirk Scherler[1,2], Wolfgang Schwanghart[3]

[1] GFZ German Research Centre for Geosciences, Section 3.3, Telegrafenberg, D-14473 Potsdam, Germany
[2] Institute of Geological Sciences, Freie Universität Berlin, D-14195 Berlin, Germany
[3] Institute of Environmental Sciences and Geography, University of Potsdam, D-14473 Potsdam, Germany

*Correspondence to*: Dirk Scherler (scherler@gfz-potsdam.de)

**Abstract.**

We propose a novel way to measure and analyse networks of drainage divides from digital elevation models. We developed
an algorithm that extracts drainage divides, based on the drainage basin boundaries defined by a stream network. In contrast
to streams, there is no straightforward approach to order and classify divides, although it is intuitive that some divides are more
important than others. A meaningful way of ordering divides is the average distance one would have to travel down on either
side of a divide to reach a common stream location. However, because measuring these distances is computationally expensive
and prone to edge effects, we instead sort divide segments based on their tree-like network structure, starting from endpoints
at river confluences. The sorted nature of the network allows assigning distances to points along the divides, which can be
shown to scale with the average distance downslope to the common stream location. Furthermore, because divide segments
tend to have characteristic lengths, an ordering scheme in which divide orders increase by one at junctions mimics these
distances. We applied our new algorithm to the Big Tujunga catchment in the San Gabriel Mountains of southern California
and studied the morphology of the drainage divide network. Our results show that topographic metrics, like the downstream
flow distance to a stream, and hillslope relief attain characteristic values that depend on the drainage area threshold used to
derive the stream network. Portions along the divide network that have lower than average relief or are closer than average to
streams are often distinctly asymmetric in shape, suggesting that these divides are unstable. Our new and automated approach
thus helps to objectively extract and analyse divide networks from digital elevation models.

# 1 Introduction

Drainage divides are fundamental elements of the Earth's surface. They define the boundaries of drainage basins and thus form barriers for the transport of solutes and solids by rivers. It has long been recognized that drainage divides are not static through time, but that they are mobile and migrate laterally (e.g., Gilbert, 1877). The lateral migration of divides is a consequence of spatial gradients in surface uplift (positive or negative) as well as stream captures. These frequently accompany tectonic deformation due to shearing, stretching, and rotating stream networks (Bonnet, 2009; Castelltort et al., 2012; Goren et al., 2015; Forte et al., 2015; Guerit et al., 2018), but recent studies have shown that even in tectonically inactive landscapes, drainage divides migrate over prolonged periods of time (Beeson et al., 2017). Such behaviour is consistent with the notion that small and local perturbations can trigger nonlocal responses with potentially large effects on drainage form and area (Fehr et al., 2011; O'Hara et al., 2019). At regional scales, mobile divides can lead to profound changes in drainage configurations and subsequent alterations of base-level and sediment dispersal to sedimentary basins. For example, Cenozoic building of the eastern Tibetan Plateau margin has been proposed to account for major reorganization of large East Asian river systems and associated changes in sediment delivery to marginal basins (Clark et al., 2004; Clift et al., 2006). Moreover, changes in drainage area that are associated with migrating divides affect river incision rates (Willett et al., 2014), and thus the topographic development of landscapes, which potentially confounds their interpretation in the context of climatic and tectonic changes (Yang et al., 2015).

Recent studies of the causes and effects of mobile drainage divides focused on topographic differences across several specific, manually selected drainage divides (e.g., Willett et al., 2014; Goren et al., 2015; Whipple et al., 2017; Buscher et al., 2017; Beeson et al., 2017; Gallen, 2018; Guerit et al., 2018; Forte and Whipple, 2018). Even if appropriate in these studies, such a procedure introduces unwanted subjectivity, both in the selection of divides and how any across-divide comparison is done. This choice of procedure may be attributed to the fact that, so far, no straightforward approach exists to reliably extract the drainage divide network from a digital elevation model (DEM). Functions that classify topographic ridges (the common shape of drainage divides) based on local surface characteristics and a threshold value (e.g., Little and Shi, 2001; Koka et al., 2011) are prone to misclassifications. The grey-weighted skeletonization method by Ranwez and Soille (2002) (homotopic thinning) requires the determination of topographic anchors (e.g., regional maxima), which makes it sensitive to DEM errors. The approach by Lindsay and Seibert (2013), who identified pixels belonging to drainage divides based on confluent flow paths from adjacent DEM pixels and a threshold value, is computationally expensive and sensitive to edge effects that depend on DEM size. Furthermore, drainage divides that coincide with pixel centres are inconsistent with the commonly used D8 flow routing algorithm (O'Callaghan and Mark, 1984), in which each pixel belongs to a specific drainage basin. A probabilistic approach, based on multiple flow directions, exists (Schwanghart and Heckmann, 2012), but computation is expensive and thus restricted to a few drainage basin outlets. Finally, all of these approaches merely yield a classified grid, but no information about the tree-like network structure of drainage divides, which requires ordering of the divide pixels into a network (Figure 1).

Although divide networks might be thought of as mirrors of stream networks, there exist fundamental differences between the two. Starting at channel heads, i.e., the tips of stream networks, streams always flow downhill and the upstream area monotonically increases. Stream networks are therefore directed networks that have a tree-like structure, and a natural order, which has been quantified in different ways (e.g., Horton, 1945; Strahler, 1954; Shreve, 1966). Divide networks, however, are neither directed nor rooted, and they may even contain cycles. They do not obey any monotonic trends in elevation, or other topographic properties that could be easily measured. As a consequence, their ordering is less straightforward. Nevertheless, it is intuitive that some divides (e.g., a continental divide) should have a different order than others. In addition, the structure of divide networks could be important in their susceptibility to drainage captures. For example, higher-order divides may record perturbations longer, as they are farther away from the base level and thus cannot adjust as quickly as lower-order divides. Furthermore, where higher-order divides are close to higher-order streams, drainage-capture events would result in profound changes in drainage area and thus a greater impact on stream discharge and power (e.g., Willett et al., 2014).

In this study, we propose to measure and analyse networks of drainage divides to address questions like: How is the geometry of a divide network related to that of a stream network? Do similar scaling relationships apply? And: Can the divide network be used to infer catchment/drainage dynamics? Empirically driven answers to these questions require tools to study drainage divides, most efficiently from DEMs. We present our study in two separate papers. In the following, part 1, we present a new approach that allows identification and ordering of drainage divides in a DEM. We investigate ways of ordering drainage divide networks and analyse basic statistical as well as topographic properties with a natural example from the Big Tujunga catchment in the San Gabriel Mountains in Southern California. In part 2 of this study (Scherler and Schwanghart, 2020), we present the results from numerical experiments with a landscape evolution model that we conducted to examine the response of drainage divide networks to perturbations.

## 2    Theoretical considerations

### 2.1    Drainage divides in digital elevation models

Drainage divides are the boundaries between adjacent drainage basins and thus their determination is based on the definition of drainage basins. In a gridded DEM, drainage basins are generally defined through the use of flow direction algorithms. The D8 flow direction algorithm (O'Callaghan and Mark, 1984) assigns flow from each pixel in a DEM to one of its eight neighbours in the direction of the steepest descent. As a result, each pixel is associated with a distinct upstream, or uphill, drainage basin. In contrast, multiple flow direction algorithms, such as the D$\infty$ flow direction algorithm (Tarboton, 1997), split the flow from one pixel to several others, which results in some pixels contributing to more than one drainage basin (Schwanghart and Heckmann, 2012). In the following, we only consider drainage basins derived from the D8 flow direction algorithm. In gridded DEMs, flowpaths derived from this algorithm run along pixel centers (Armitage, 2019), and there exists the possibility that two flowpaths run parallel to each other in neighboring pixels. As a consequence, drainage basin boundaries, and thus divides, must be located in between DEM pixels, and have infinitesimal width (Fehr et al., 2009). Another important

consequence is that divides will have only two possible orientations that are parallel to pixel boundaries. Our definition of divides is different from one in which divides are linked to the highest points (pixels) on interfluves (Haralick, 1983). In the case of multiple flow directions, for example, a meaningful position of a drainage divide would be the place within a pixel that partitions the pixel area according to the flow contributions to adjacent drainage basins.

For a given point in a channel network, its drainage basin is uniquely defined to be the upstream area of that point. The drainage divide of that basin, however, does not intersect the channel itself. We thus define drainage divides as lines (or graphs) that mark the margin of drainage basins and that do not cross rivers (Figure 2). When derived from a DEM, these graphs consist of nodes and edges: nodes are located on pixel corners and edges follow pixel boundaries. A meaningful property of divide nodes and edges is that they should not coincide with nodes or edges of the drainage network. When applying the D8 flow routing

algorithm to a gridded DEM, with square elevation cells, however, this requirement poses a problem, due to the different pixel connectivity of divides and rivers. Whereas divide nodes can be connected to only four cardinal neighbours, river nodes can be connected to eight different neighbours. In consequence, divide nodes may exist that coincide with diagonal edges of drainage networks (Figure 2). In a gridded DEM this issue could be resolved with a D4 flow direction algorithm; or, more generally, this issue could be avoided if flow is only allowed orthogonal to cell boundaries. In our approach, we nonetheless

adopt the D8 flow direction algorithm and allow for spatial congruence of streams and divides. In practice, such issues mainly arise near confluences (Figure 2).

## 2.2     Drainage divide networks

Analogous to streams, drainage divides are typically organized into tree-like networks (Figure 1), although cycles that correspond to internally-drained basins may exist. Because of the directed flow of water, stream networks can be regarded as

directed graphs that start at channel heads (the leaves of the tree) and end at an outlet or a river mouth (the root of the tree). Flow directions in stream networks can be easily derived from node elevations (e.g., O'Callaghan and Mark, 1984) and the hierarchy of streams can be related to their upstream area, for example. In contrast, drainage divides have no inherent direction, and there exists no terrain property, like elevation, that could be used to assign a direction to them. A meaningful metric for ordering divides may be the average *branch length* (Lindsay and Seibert, 2013), i.e., the average distance $\Lambda$ [m] one would

have to travel down on either side of a divide to reach a common stream location (Figure 3). However, measuring this distance requires that the common stream location and the entire path leading to it be contained in the DEM. Because this may not be true for a significant part of the divide network in a DEM and because measuring this distance is computationally expensive (Lindsay and Seibert, 2013), it is not very practical.

Instead, we suggest that directions can be derived from the tree-like structure of drainage divide networks. Analogous to a

120 parcel of water that travels down a river from its source to its mouth, we propose to start at the leaves of the tree, which we call the *endpoints* of the divide network (Figure 2), and incrementally move down the branches (Figure 4). Note that the term "move down" does not refer to elevation, but to the hierarchy of the divide network. Where two or more drainage divides meet, they form a *junction*. We call individual parts of drainage divides that link endpoints and junctions, junctions and junctions, or

endpoints and endpoints, the drainage-divide *segments*, and we refer to the ends of divide segments as *segment termini* to avoid confusion with endpoints. At junctions with more than one unsorted divide segment, the sorting process pauses because it is not obvious in which direction the sorting shall continue. However, in the absence of cycles (internally drained basins), each junction will reach a point in the sorting loop when there is only one unsorted divide segment left so that the sorting can continue (Figure 4). This condition assures that the divide segments are correctly sorted in a tree-like manner, but it fails when encountering a cycle. As we will show later, the average branch length $\Lambda$ scales linearly with the maximum distance from an endpoint along the sorted divide network as well as the maximum number of divide segments (or junctions), both of which are more easily computed. We thus propose to order the nodes and edges of the divide network, by their maximum distance from divide endpoints, measured either in map units or in the number of divide segments. From now on, we call the distance measured in map units along the directed divide network the divide distance ($d_d$).

## 3    Materials and Methods

### 3.1    Divide algorithm

We implemented the above-described way of extracting and ordering drainage divides from a DEM in the TopoToolbox v2 (Schwanghart and Scherler, 2014), a MATLAB-based software for topographic analysis. Figure 5 shows the workflow of our approach, which consists of the following steps:

(1)    For a given DEM, we first define a stream network, based on the D8 flow direction algorithm and a threshold drainage area (Figure 5a,b). The lower the threshold, the more detailed the stream and divide networks will be.

(2)    We extract drainage divides, based on drainage basin boundaries that we obtained for drainage areas at tributary junctions and drainage outlets (Figure 5c). Initially, each drainage basin boundary is composed of one divide segment that connects two endpoints, and junctions do not yet exist. These divide segments do not cross any rivers but their nodes may coincide with stream edges (Figure 2). We remove redundant divide segments in the collection of divides, which arise from nested and adjoining drainage basins. As a result, we are left with a set of unique divide segments, which, however, may be continuous across junctions or terminate where they should be continuous (Figure 6).

(3)    We next organize the collection of divide segments into a drainage divide network (Figure 5d). This is the core of the algorithm, in which we identify endpoints and junctions, merge broken divide segments, and split divide segments at junctions (Figure 6). Our algorithm distinguishes between junctions, endpoints, and broken divide segments, by computing for each node of the divide network the number of edges linked to it, the number of segment termini linked to it, and the existence and direction of a diagonal flow direction. For example, most nodes with two edges and two segment termini correspond to a broken segment and need to be merged, unless they coincide with a stream and merging them would make the resulting divide cross that stream (Figure 6). See Appendix 7.1 for more details.

(4) Finally, we sort the drainage divide segments within the network (Figure 5e). The algorithm iteratively identifies segments that are connected to endpoints and removes them from the list of unsorted divide segments until no divide segments are left (Figure 4). This step assigns a direction to each divide segment and transforms the divide network into a directed acyclic graph. For the sorted divide network, we then compute the divide distance, i.e., the maximum distance from an endpoint along the sorted divide network (Figure 5f).

After the sorting, we also assign orders to divide segments, based on the ordering of stream networks, first introduced by Horton (1945). We adopted both Strahler's (1954) and Shreve's (1966) rules of stream ordering, and added a third rule that we call *Topo*. All ordering schemes start with a value of one at endpoints and progressively update divide orders at junctions based on the following rules:

$$Strahler: \omega_k = \max(\min\{\omega_i\} + 1; \omega_i) \qquad \text{Eq. 1}$$

$$Shreve: \omega_k = \sum_{i=1}^{n} \omega_i \qquad \text{Eq. 2}$$

$$Topo: \omega_k = \max(\omega_i) + 1 \qquad \text{Eq. 3,}$$

where $\omega_i$ are the divide orders of the $n$ joining divide segments and $\omega_k$ is the divide order of the following divide segment. In the *Strahler* ordering scheme, the order increases by one if the joining divide segments have the same order, otherwise it remains at their maximum order. In the *Shreve* ordering scheme, the resulting divide order is the sum of those of the joining divide segments, and in the *Topo* ordering scheme, divide orders increase by one at each junction. Junctions typically link three different divide segments, but up to four can occur (Figure 6). Based on the *Strahler* ordering scheme, the bifurcation ratio $R_b$ can be derived from (Horton, 1945):

$$R_b = \frac{N_\omega}{N_{\omega-1}} \qquad \text{Eq. 4,}$$

where $N_\omega$ is the number of divide segments of order $\omega$.

As aforementioned, our divide algorithm currently does not handle internally drained basins. Whereas the divides of internally drained basins are easy to identify, they are not easily sorted in a meaningful manner. In fact, the distance to a common stream location ($\Lambda$) is undefined for a divide of an internally drained basin. At the moment, the sorting procedure (Figure 4) stops at such divides, because the divide segments cannot be assigned a direction. In consequence, also parts of the divide network that potentially lie beyond an internally-drained basin, and for which the distance to a common stream location is defined, cannot be reached anymore. While we are working on a solution to this issue, our algorithm is currently best applied to acyclic drainage divide networks.

## 3.2 Topographic data and analysis

We investigated basic characteristics of drainage divide networks using a 30-m resolution DEM from the 1-arc second Shuttle Radar Topography Mission data set (Farr et al., 2007). We focused on the catchment of the Big Tujunga River in the San Gabriel Mountains, USA. The catchment is a good example of a transient landscape with active drainage basin reorganization

and landscape rejuvenation as the river incises into a relict pre-uplift landscape (DiBiase et al. 2015). We pre-processed the DEM by carving through local sinks (Soille et al. 2003) to avoid artificial internally drained basins, and we obtained a stream network based on a minimum upstream area of 0.1 km$^2$. We note that this threshold is likely well within the zone of debris flow instead of fluvial incision (Stock and Dietrich, 2003), but for the purpose of our analysis, this is irrelevant.

We analysed the divide network, its planform geometry, and its relation to topography. Planform geometry is studied using statistical analysis of the number and length of divide segments of different orders. Topographic analyses are based on metrics that we determined for the entire DEM and that we subsequently associated, or mapped, to divide edges and entire divide segments. As topographic metrics, we focus on hillslope relief (*HR*) and horizontal flow distance to the stream network (*FD*). *HR* was defined to be the elevation difference between a point on the divide and the point on the river that it flows to. To quantify the morphologic asymmetry of a divide, we propose to use the across-divide difference in hillslope relief ($\Delta HR$), normalized by the across-divide sum in hillslope relief ($\sum HR$), and call its absolute value the divide asymmetry index (*DAI*):

$$DAI = \left| \frac{\Delta HR}{\sum HR} \right| \qquad\qquad \text{Eq. 5.}$$

The *DAI* ranges between 0 for entirely symmetric divides and 1 for the most asymmetric divides. Note that this index is based only on values of hillslope relief (*HR*). Theoretically, a divide with equal amounts of *HR* on either side of a divide, but contrasts in flow distance (*FD*) and thus slope angle, would yield a *DAI* of zero. However, due to the definition of streams by a minimum drainage area, this hardly ever occurs. In addition, such cases can be identified by cross-divide differences in *FD*.

## 4    Results

### 4.1    Basic divide statistics

We applied our divide algorithm to the Big Tujunga catchment, and the resulting divide network for different ordering schemes is shown in Figure 7. Because the *Shreve* and *Topo* ordering schemes yield larger ranges in divide orders, their visualization allows for greater differentiation compared to the *Strahler* ordering scheme. Differences in the visual appearance of the divide network due to the ordering scheme are also apparent at the root node, i.e., the junction that is encountered last in the ordering process (black arrows in Figure 7). In the *Topo* ordering scheme, divide orders increase by one during each sorting cycle, so that the last divide segments will have orders that are different by not more than one. In contrast, the ordering rules of the *Strahler* and *Shreve* scheme (see Eq. 1 and Eq. 2) may yield unequal orders during the sorting, so that the divide orders of the last divide segments may be different by more than one. In the Big Tujunga catchment, the basin area and thus the number of divide segments is larger north of the Big Tujunga River, compared to south of it. As a consequence, both *Strahler* and *Shreve* divide orders increase more rapidly along the northern perimeter compared to the southern, and the junction encountered last during the sorting process (at the root of the tree) opposes divide segments with orders of 7 and 6 for *Strahler*, and 1463 and 772 for *Shreve*, in the north and south, respectively (Figure 7). For the *Strahler* ordering scheme, the frequency distribution of divide segments decreases exponentially with divide order ($\omega$), which is consistent with Horton's law of stream numbers

(Horton, 1945), and corresponds to a bifurcation ratio of $R_b = 3.89 \pm 0.30$ (standard error). The bifurcation ratio of the associated stream network is $5.39 \pm 0.87$.

We computed divide segment lengths for different drainage area thresholds ($A_{min}$) and show the associated empirical distribution functions in Figure 8a. Divide segment lengths are not normally distributed, but can be reasonably fitted with a gamma distribution. However, the fitted gamma distributions predict systematically higher probabilities for shorter divide segments and lower probabilities for longer divide segments compared to the actual data. For the different drainage area thresholds that we tested, the shape parameter of the fitted gamma distribution ($k$) attains values that range between 0.87 and

1.75. In general, the average length of all divide segments increases with the drainage area threshold used for deriving the stream network, simply because both the stream and the divide network extend to finer branches. For a drainage area threshold of 0.1 km$^2$, the average length across all divide orders is $442 \pm 323$ m ($\pm 1\sigma$), compared to an expected value (= $k\theta$) of ~442 from the fitted gamma distribution. The average length for different divide orders tends to be slightly lower at $\omega < \sim 40$ (based on the *Topo* ordering scheme), compared to $\omega \geq \sim 40$ (Figure 8b).

We quantified the average *branch length*, i.e., the average distance one would have to travel down on either side of a divide to reach a common stream location ($\Lambda$), for 100 randomly chosen divide edges per divide order in the *Topo* ordering scheme. Although the maximum order for which $\Lambda$ can be determined (because of the size of our DEM) is limited to $\omega \leq 55$, results demonstrate that $\Lambda$ (in km) increases linearly as $0.36 \times$ divide order ($\omega$), and $1.11 \times$ divide distance ($d_d$) (Figure 9). The linear scaling of these two relationships is a consequence of the similarity of segment lengths for different divide orders (Figure 8b).

Whereas the *Topo* ordering scheme can be approximated by dividing the divide distance by the expected divide segment length, this does not hold true for the *Strahler* and *Shreve* ordering schemes, which yield relationships between $\Lambda$ and divide order that are non-linear (not shown).

## 4.2 Drainage divide network morphology of the Big Tujunga catchment

We next studied the morphology of the drainage divide network from the Big Tujunga catchment. Because the divide
morphology consists of parts that lie within the catchment as well as parts that lie outside of it, we analysed the entire drainage divide network from the DEM as shown in Figure 5. Although the drainage divide network is truncated along the DEM edges, the following analysis is insensitive to this issue. Figure 10 shows the drainage divide morphology of the Big Tujunga catchment, based on a stream network that was derived from a drainage area threshold of 1 km$^2$. Topographic metrics are shown for each divide edge (Figure 1). Whereas across-divide mean flow distance ($\overline{FD}$) varies between 0 and ~3000 m, mean
hillslope relief ($\overline{HR}$) varies between 0 and ~800 m. It is notable that the biggest range in values occurs at divide distances <10 km, whereas divides at higher distances appear to hover around characteristic values that are controlled by the drainage area threshold used to derive the stream network (Figure 10e). It should be noted that divide edges at low divide distances are much more abundant compared to those at higher distances, or equivalently, at higher divide orders (Figure 7). At increasingly higher drainage area thresholds however, the frequency of divides at low order and distance decreases more rapidly compared to the
frequency of high orders and distances. The average ($\pm 1\sigma$) $\overline{FD}$ and $\overline{HR}$ values for a drainage area threshold of 1 km$^2$ are 1325

± 350 m and 341 ± 129 m, respectively. The empirically determined average $\overline{FD}$ values for all tested drainage area thresholds are consistent with Hack's law (Hack, 1957), which relates the length $L$ of the longest stream in a catchment to its drainage area $A$, according to $L = k_a A^h$. Values of $k_a$ and $h$ are ~1.6 and ~0.5, respectively – similar to values observed elsewhere (Hack, 1957). Combining $\overline{HR}$ and $\overline{FD}$ yields average slope values that vary between ~20° and ~8°, for drainage area thresholds between 0.1 km² and 10 km². The mean slope value of the entire DEM is ~21.5°, which suggests that the lower drainage area threshold better confines the divides to hillslopes as compared to lower-sloping channels.

Based on the observation of characteristic values of $\overline{FD}$ and $\overline{HR}$, we sought to identify parts of the divide network that have anomalously low relief or are anomalously close to a stream. Instead of mean values, we turned towards across-divide minimum values of flow distance ($FD_{min}$) and hillslope relief ($HR_{min}$), as these would be more sensitive to deviations on either side of a divide. In addition, we compared these values with the divide asymmetry index ($DAI$), as we expected that anomalous divides may also be topographically asymmetric (e.g., Whipple et al., 2017). Figure 11 shows how $HR_{min}$, $FD_{min}$, and $DAI$ vary with distance along the divide network of the Big Tujunga catchment. Notable deviations from average values (Figure 10) occur at ~22-25 km, ~ 41-45 km, and >54 km divide distance, and are typically associated with asymmetric divides (Figure 11). Highly asymmetric divides are furthermore found at low divide distances (<5 km) and typically coincide with low values of $HR$, whereas $FD$ could be high or low. The systematic decrease in $FD$ and $HR$, concurrent with an increase in the $DAI$ at higher divide distances prompted us to query the geographic position of these divides and how they compare to the surrounding landscape. We thus imposed thresholds to identify anomalously low ($HR_{min}$ <200 m) and asymmetric divides ($DAI$>0.5), which are less than 1000 m from a stream ($FD_{min}$ <1000m) (Figure 12). Beheaded streams as well as sharp-crested and shortened hillslopes identified in high-resolution satellite imagery (Figure 12b-e) support the impression that these divides are mobile and migrating in the direction of lower $HR$ and sometimes shorter $FD$. Most of these divides can be seen to border regions of contrasting local relief (Figure 12a), and many cluster along the eastern edge of the catchment. The predicted migration direction indicated in Figure 12a is derived from the orientation of the divide segments and their mean $DAI$ magnitude. If correct, most of the divide migration along the southern and eastern edge of the catchment, from higher to lower relief, would result in area loss for the Big Tujunga catchment.

## 5    Discussion

### 5.1    Extraction and ordering of drainage divide networks

Our new approach allows routinely extracting drainage divides from any DEM without internally drained basins. We have shown that the maximum divide distance $d_d$, calculated as the maximum distance along the (directed) divide network from an endpoint, is a meaningful metric for ordering drainage divide networks, as it scales linearly with the average branch length $\Lambda$, i.e., the average distance one would have to travel down on either side of a divide to reach a common stream location (Figure 9). In contrast to the average branch length, however, the divide distance is more easily and rapidly calculated and is less prone

to edge effects that inhibit the ordering of divides (Lindsay and Seibert, 2013). However, whenever a drainage basin intersects the edge of a DEM, its truncation will likely produce a spurious drainage divide. Furthermore, calculated divide distances are most likely lower than they would be for a larger DEM, similar to the reduction of upstream area along a stream network. Truncated drainage basin boundaries should therefore be avoided, or discarded from analyses that rely on correct divide distances.

The proposed sorting procedure (Figure 4) recovers the tree-like structure of the divide network and allows the derivation of divide orders, analogous to the well-known stream orders. Because divide segments have similar mean lengths across all divide orders (Figure 8), divide orders derived with the *Topo* ordering scheme can serve a similar purpose as divide distance. Shreve (1969) studied link lengths in stream networks and concluded that their distribution is better described with a gamma distribution compared to an exponential or log-normal distribution. Results from the Big Tujunga catchment support this conclusion with respect to divide segment lengths, although systematic deviations can be observed (Figure 8a). It needs to be tested with more observations whether these deviations are inherent to drainage divide networks in general, and whether they could hold clues about the dynamic state of a landscape.

An advantage of characterizing the divide network by distance instead of orders is that the divide distance is invariant with respect to the chosen drainage area threshold, whereas divide orders are not, because they depend on the total number of divide segments and junctions. Further differences are apparent at the root node, which may oppose divide segments with orders that differ by more than one (Figure 7). In the case of the Big Tujunga catchment, *Strahler* orders are not that different across the root node, but in a different landscape that could well be the case. This issue is more prevalent in the case of *Shreve* ordering, but it is avoided with the *Topo* ordering scheme. Furthermore, the non-uniform distribution of divide segment lengths (Figure 8) influences how similar or dissimilar divide distances of the meeting divide segments are at the root node. If the average divide segment length of trees that meet at the root node are different, divide distances will make a jump, even if divide orders are similar. In the Big Tujunga catchment, the divide distance jump at the root node is 5400 m.

Divide orders derived with the *Strahler* ordering scheme can be used to investigate how the divide network conforms to Horton's (1945) laws of network composition. In the Big Tujunga catchment, for example, the bifurcation ratio of the divide network ($R_b$ ~3.9) is lower than that of the stream network ($R_b$ ~5.4). This may in part be due to the fact that we analysed only a part of the divide network; divide segments that originate from the main catchment boundary and extend outwards are not included in the statistics. Including those in the calculation yield $R_b$ ~4.6 for the divide network, still lower than the bifurcation ratio of the stream network. Nevertheless, these bifurcation ratios are similar to published bifurcation ratios of different natural stream networks (e.g., Tarboton et al., 1988), supporting the expected similarity of the stream and divide network topology. However, more observations from different landscapes are needed to assess systematic differences and commonalities between divide and stream networks.

## 5.2    Drainage divide mobility in the Big Tujunga catchment

Based on the observation of characteristic values of minimum hillslope relief (300-500 m) and minimum flow distance (1000-1800 m), we identified drainage divides in the Big Tujunga catchment that are anomalously low, close to a stream, and asymmetric (Figure 11, Figure 12). These geometric properties suggest the existence of windgaps, hillslope undercutting by rivers, and spatial anomalies in erosion rates, which are diagnostic for past or ongoing mobility of drainage divides. Anomalous drainage divides are particularly frequent along the eastern edge of the catchment, where an area of low hillslope angles and local relief (Figure 12), the so-called Chilao Flats, is bordering a steep catchment to the south and east of it. This high-elevation low-relief area is thought to represent a relict peneplain surface that has been uplifted during growth of the San Gabriel Mountains, and is currently destroyed by headward incision of rivers (Spotila et al., 2002; DiBiase et al., 2015). Cosmogenic [10]Be-derived erosion rates confirm lower erosion rates in the Chilao Flats area compared to the surrounding steeper catchments (DiBiase et al., 2010), which ought to drive divide migration and drainage area loss in the headwaters of the Big Tujunga catchment, consistent with our results.

We identified another stretch of anomalous divides along the southern margin of the Big Tujunga catchment (Figure 12a, d), part of which is coincident with the trace of the San Gabriel Fault, which follows the orientation of the valley (Morton and Miller, 2006). Reduced relief in a ~1-km wide zone around this fault is also observed farther to the east, along the West Fork of the San Gabriel River (Scherler et al., 2016), suggesting weaker rocks closer to the fault (e.g., Roy et al., 2015). Other anomalous divides in this area, as well as along the northern margin of the Big Tujunga catchment, show signs of mobility by one-side shortened hillslopes and beheaded valleys (Figure 12b, d). We thus suggest that most, if not all of the anomalous divides we identified based on hillslope relief, flow distance, and divide asymmetry, are in fact unstable and migrating with time. Because most of the peripheral divides indicate drainage area loss of the Big Tujunga catchment, these area changes ought to result in changes in stream power (Willett et al., 2014), which complicate the interpretation of stream profile knickpoints in a tectonic framework (DiBiase et al., 2015).

## 6    Conclusions

In this study, we presented an approach to objectively extract and analyse drainage divides from DEMs. We argued that divides can be ordered in a meaningful way based on the average distance one would have to travel down on either side of a divide to reach a common stream location, and we have shown that this distance can be well approximated by the maximum along-divide distance from endpoints of the divide network, which we termed the divide distance. We have also shown that the tree-like structure of divide networks lends itself to topological analysis similar to stream networks, and we introduced an ordering scheme (*Topo*), in which divide orders increase by one at divide junctions. Because divide segments tend to have characteristic lengths, the *Topo* ordering scheme mimics the divide distance. Topographic analysis of the drainage divide network of the Big Tujunga catchment yielded characteristic values of flow distance and hillslope relief that can be shown to depend on the drainage area threshold, with which the stream network was derived. Based on these characteristic values and a minimum

divide distance of ~5 km, below which we observed large scatter, we identified divides that have anomalously low hillslope relief, are close to rivers, and are asymmetric in shape. We interpret these divides to be mobile and indicating beheaded valleys or future capture events.

## 7     Appendix

### 7.1     Classification of divide network nodes

Once the drainage divides are defined, based on the outline of drainage basins, and redundant divide segments are removed, they compose a network $D = (V,E)$ which is defined by a set of vertices V (or nodes) and a set of edges E, each of which is associated with two distinct vertices. However, D may contain some divide segments that do not end at junctions or that terminate at nodes that are neither junctions nor endpoints. To create the divide network, we have to identify divide endpoints and junctions, as well as divide segments that need to be merged or parted. We achieve this by computing for each node $x_i$ the

number of edges (1 to 4) and divide-segment termini (0 to 4) that exist in D, and identifying whether the node coincides with a stream edge (0/1) (**Table A1**). Based on these criteria, we classify nodes to be endpoints (EP), junctions (J), or broken segments (BS). In the case of nodes with three edges, three segment termini, and the presence of a stream edge, we also check which of these edges, if connected, would cross a stream, to distinguish the EP from the BS. After this classification, we are able to merge broken segments, split segments at junctions, and thus update D, which now contains all the divide segments

that compose the drainage divide network.

## 8     Code availability

The divide algorithm developed in this study has been implemented in the TopoToolbox v2 (Schwanghart and Scherler, 2014). The codes will be made available with the next TopoToolbox release, and shall be accessible through https://github.com/wschwanghart/topotoolbox.

## 9     Competing interests

The authors declare that they have no conflict of interest.

## 10     Acknowledgements

D. Scherler acknowledges funding through grant SCHE 1676/4-1 from the Deutsche Forschungsgemeinschaft (DFG).

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

## 12    Tables

Table A1: Divide node classification matrix

| Edges (#) | Segment termini (#) | Stream (0/1) | Stream crossing (0/1) | Class* |
|---|---|---|---|---|
| 1 | 1 | 0 | 0 | EP |
| 1 | 1 | 1 | 0 | EP |
| 2 | 2 | 0 | 0 | BS |
| 2 | 2 | 1 | 0 | BS |
| 2 | 2 | 1 | 1 | EP |
| 3 | 1 | 0 | 0 | EP |
| 3 | 1 | 1 | 0 | EP |
| 3 | 3 | 0 | 0 | J |
| 3 | 3 | 1 | 0 | BS |
| 3 | 3 | 1 | 1 | EP |
| 4 | 4 | 0 | 0 | J |
| 4 | 4 | 1 | 0 | BS |
| 4 | 2 | 0 | 0 | J |
| 4 | 2 | 1 | 0 | BS |

* EP = Endpoint, BS = Broken segment, J = Junction


## 13    Figure captions

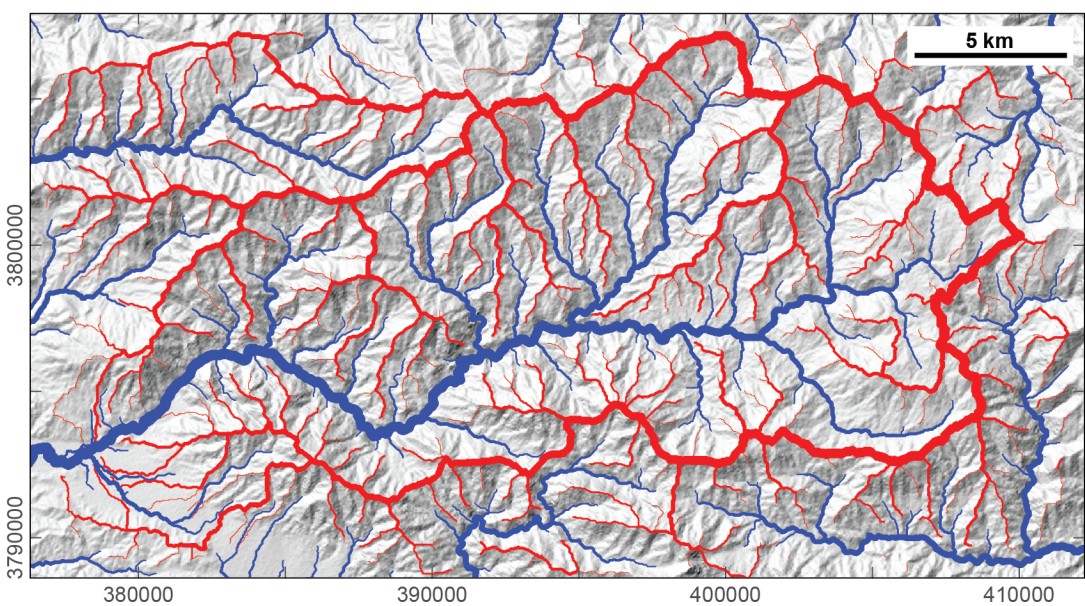

**Figure 1**: Big Tujunga catchment, San Gabriel Mountains, United States. Stream network (blue) and drainage divide network (red), draped over hillshade image. Drainage divide network obtained with the approach developed in this study. Thickness of the stream and divide lines is related to upstream area and divide order, respectively. Divide orders are based on the *Topo* ordering scheme, which we describe in the main text. The map projection is UTM zone 11. North is up.


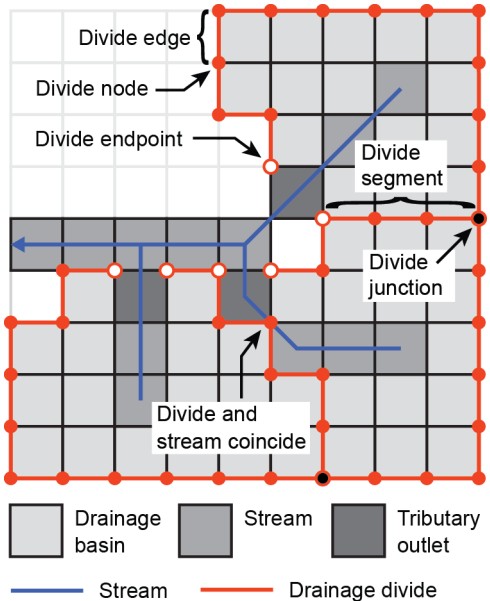

**Figure 2:** Definition of drainage divides in a digital elevation model. Note the point where a drainage divide (red) coincides with a river channel (blue). See text for details.

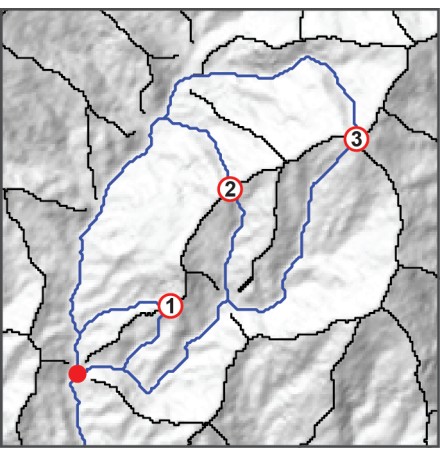

**Figure 3**: Ordering of divides based on the average distance to a common stream location, $\Lambda$. Numbered circles are places on drainage divides (black lines), and blue lines indicate the flow path to their common stream location (red circle). The resulting order of the drainage divide places is $1 < 2 < 3$. The shown landscape is part of the Big Tujunga drainage basin.

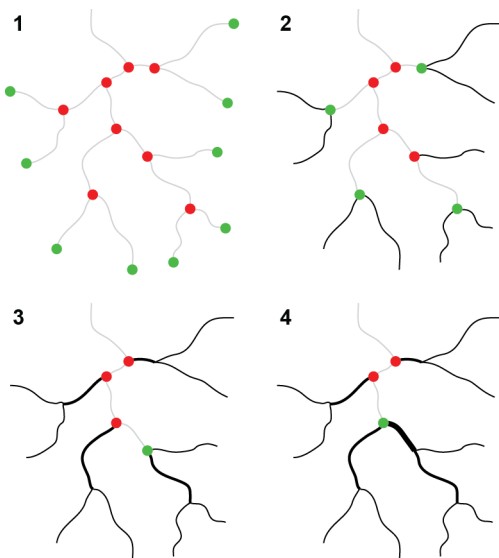

**Figure 4:** Iterative sorting and ordering of the divide network. The divide network is assembled starting with divide segments that contain endpoints (green) and which are then removed from the collection of divide segments. Former junctions (red) that have only one segment remaining become endpoints and the iteration continues until no more endpoints exist.

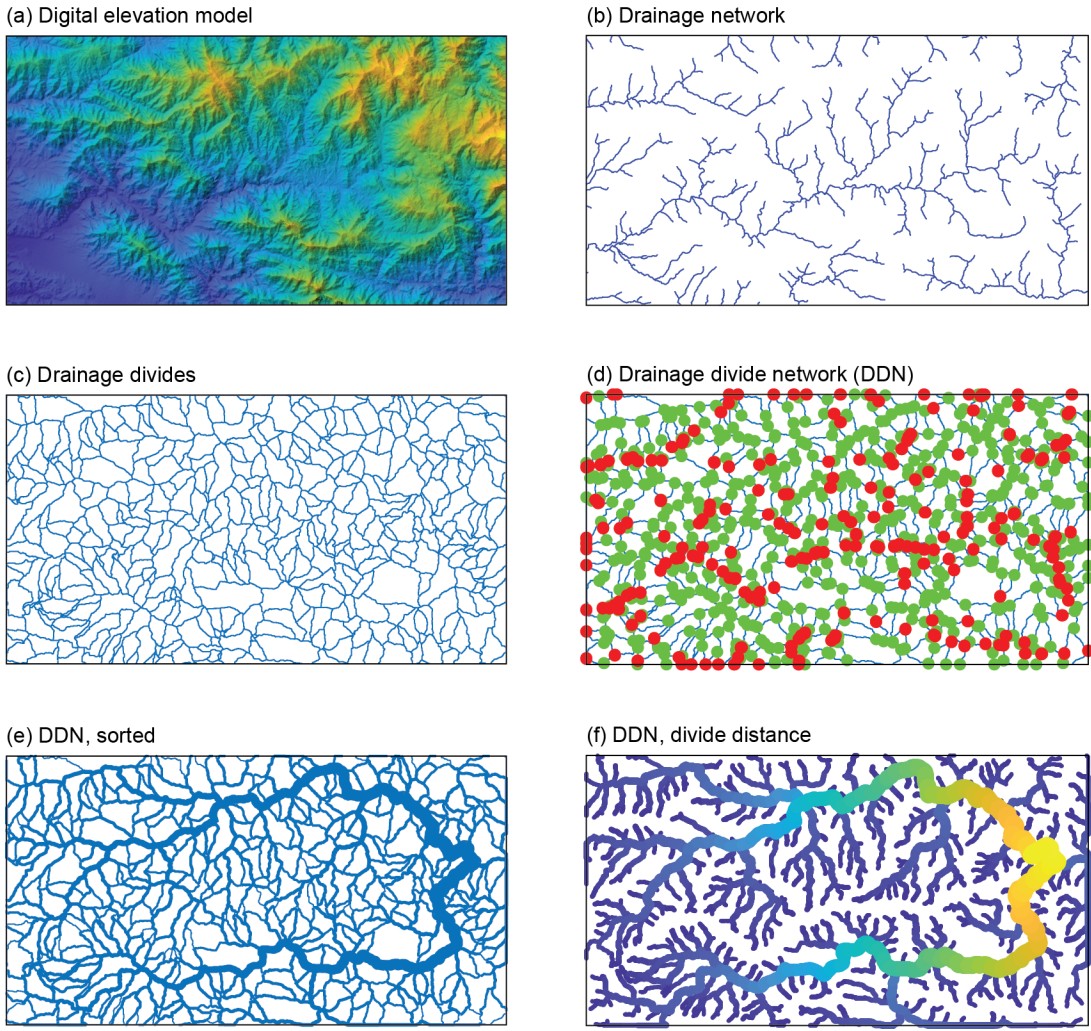

**Figure 5:** Workflow of identifying and ordering drainage divides in digital elevation models. (a) Digital elevation model of the Big Tjunga catchment, San Gabriel Mountains, California. (b) Drainage network based on a minimum upstream area of 1000 pixels (0.81 km²). (c) Drainage divides of all drainage basins upstream of confluences in b. (d) Drainage divide network with endpoints (red) and junctions (green). (e) Sorted drainage divide network. Line thickness indicates divide order from low (thin) to high (thick). (f) Drainage divide network color-coded by divide distance (blue = low, yellow = high). Note that only divides at a distance >1 km are shown.


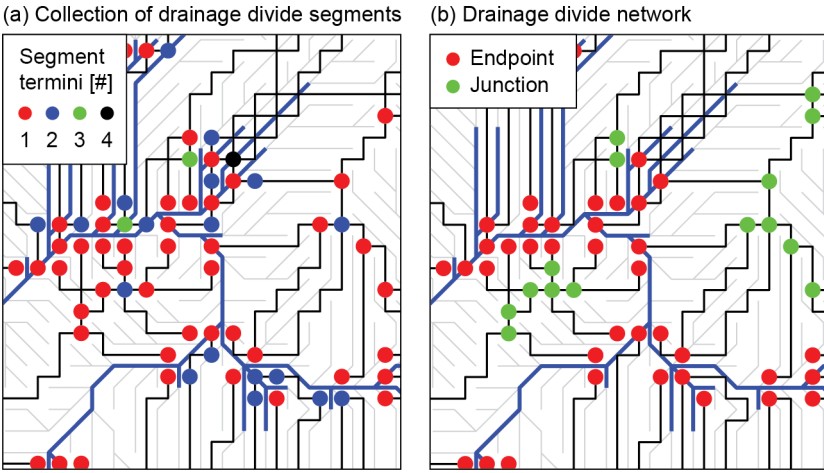

**Figure 6:** Transformation of a collection of drainage divide segments (a) into a drainage divide network with endpoints and junctions (b). Black lines are drainage divides, blue lines are streams, and flow directions are shown as light gray lines. Note that we used a minimum upstream area of only 10 pixels to define the stream network for illustration purposes.


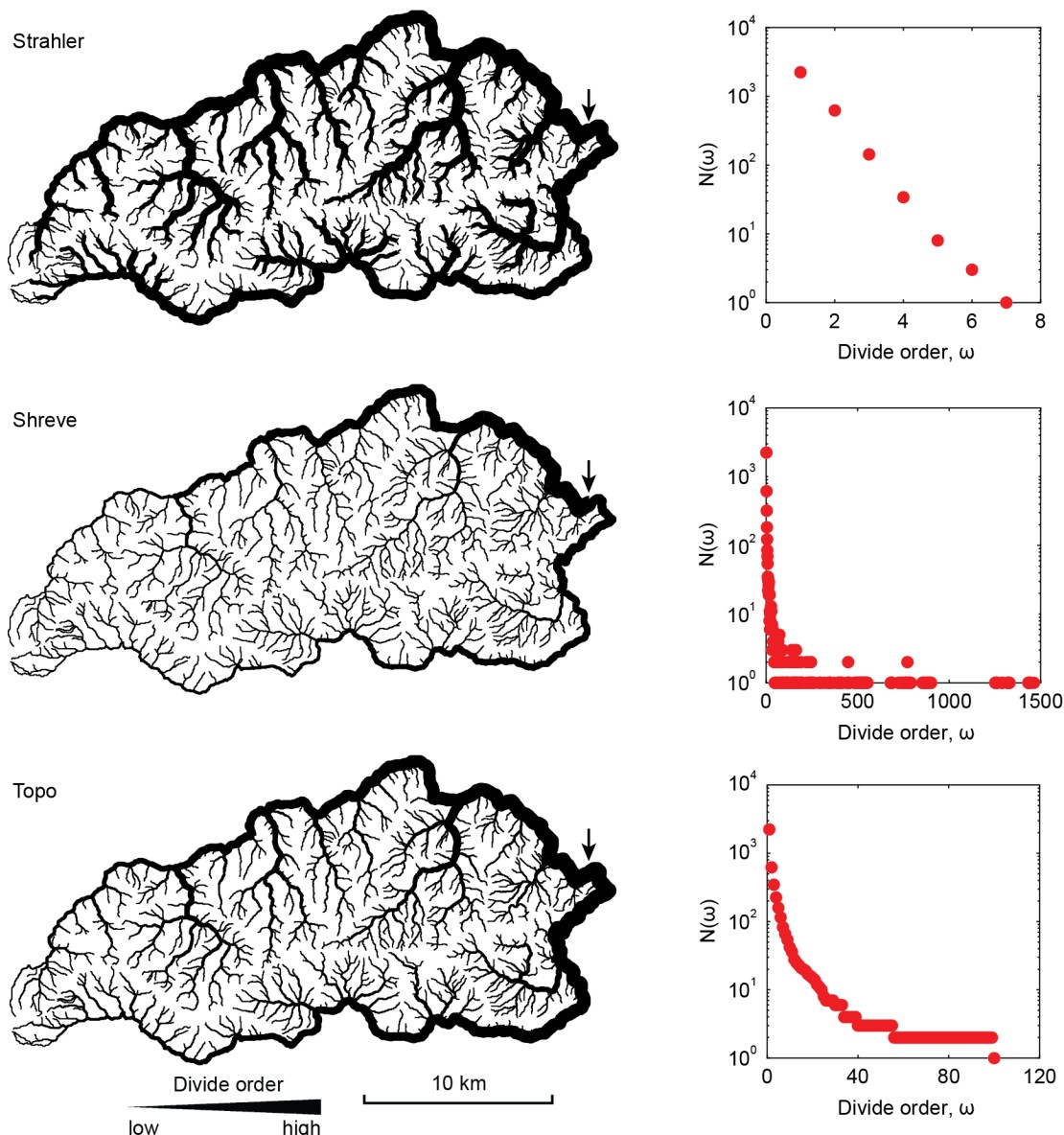

**Figure 7:** Divide network of the Big Tujunga catchment in the western San Gabriel Mountains, California, USA. Left column shows the divide network, with line thickness indicating the divide order. Arrow marks the last divide segment encountered in the sorting process, or, equivalently, the root of the tree-like network. Right column shows the number of divide segments as a function of divide order.

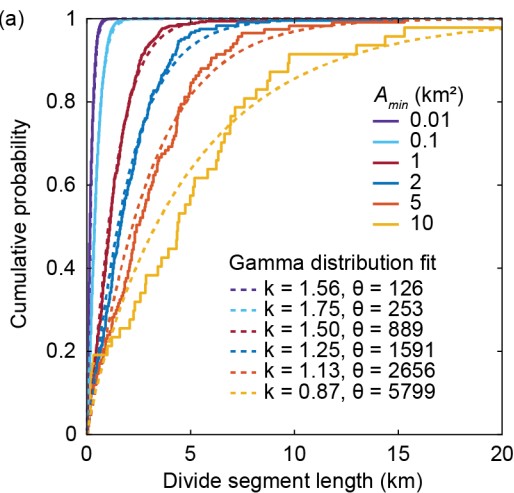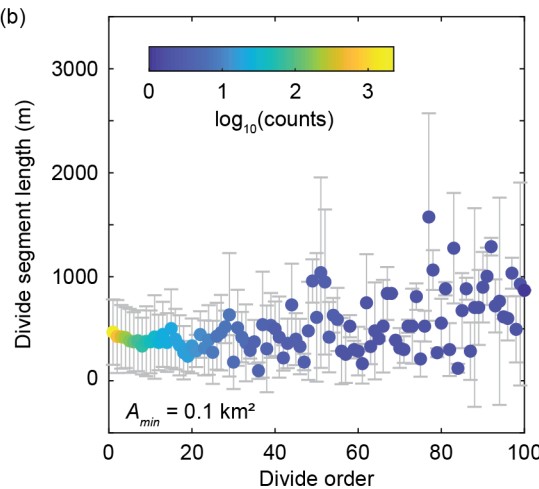

**Figure 8:** Divide segment statistics. (a) Empirical distribution functions of divide segment lengths for different drainage area thresholds ($A_{min}$), and fitted cumulative gamma distribution functions. (b) Average length ($\pm 1\sigma$) of drainage divide segments by order (*Topo* ordering scheme). The number of observations per divide order drops below 10 at an order of 25.

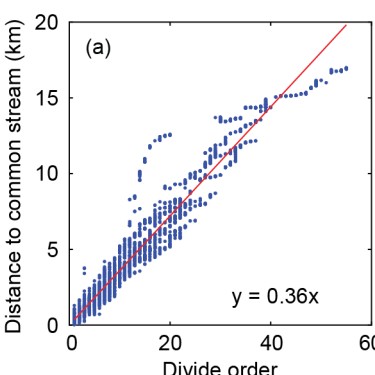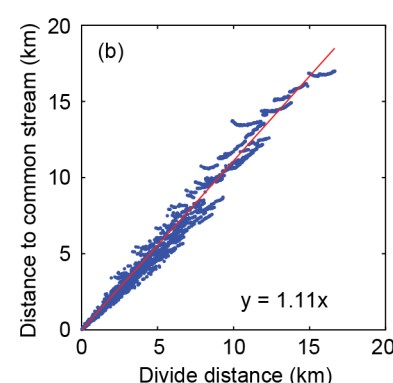

**Figure 9:** Distance to common stream location by (a) divide order and (b) divide distance, for 100 randomly chosen divide edges per divide order within the Big Tujunga catchment.

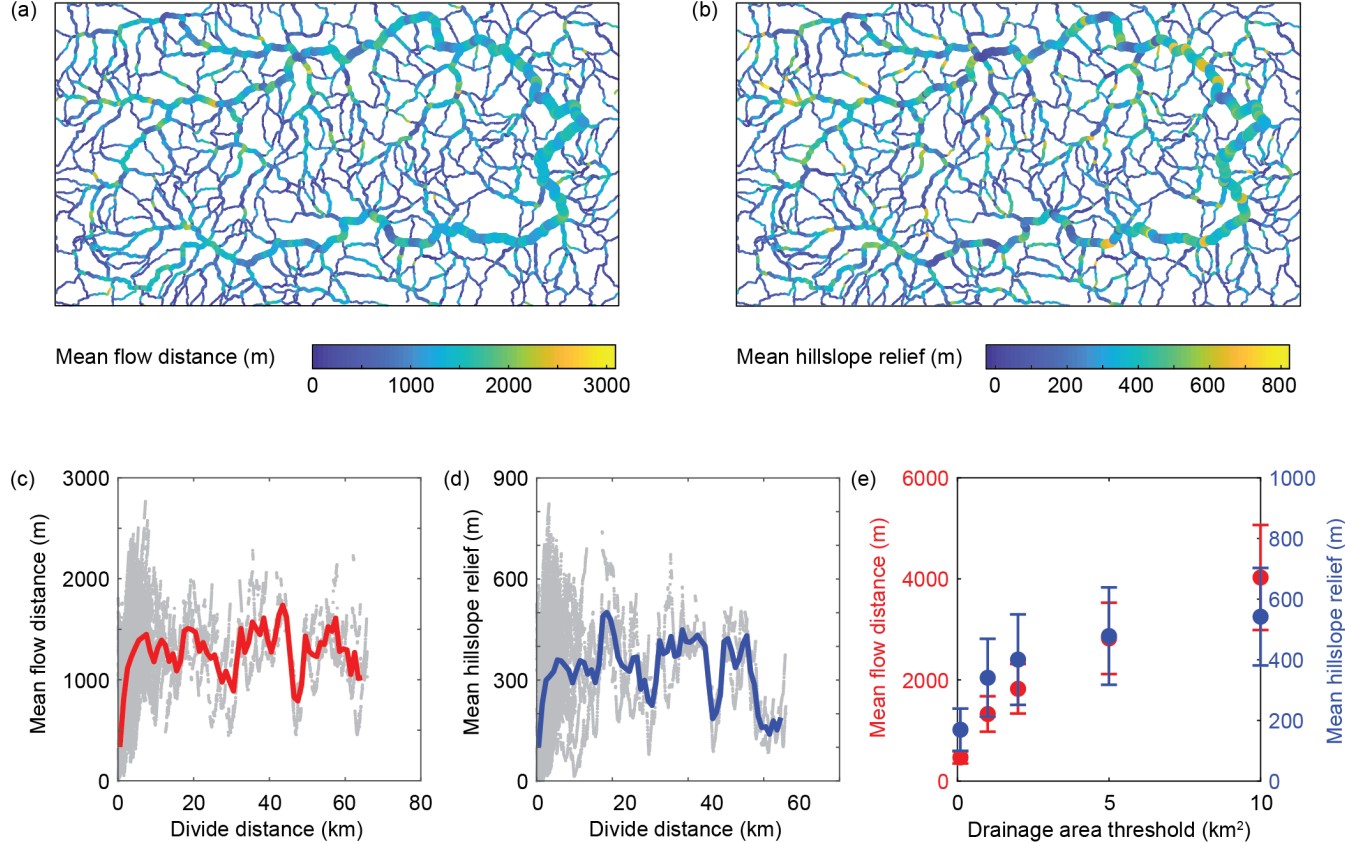

**Figure 10:** Drainage divide morphology of the Big Tujunga catchment, based on a stream network that was derived from a drainage area threshold of 1 km². (a) Drainage divide network (DDN) coloured by mean flow distance. Line thickness scales with divide distance. (b) DDN coloured by mean hillslope relief. (c) Relationship between mean flow distance and divide distance of all divide edges in (a). Red line shows 1000-m moving average. (d) Relationship between mean hillslope relief and divide distance of all divide edges in (b). Blue line shows 1000-m moving average. (e) Average (±1σ) values of mean flow distance (red) and mean hillslope relief (blue) for different drainage area thresholds. Average values were determined from all divide edges at a divide distance >5 km, to minimize the influence of divides that are close to streams simply due to their proximity to confluences.

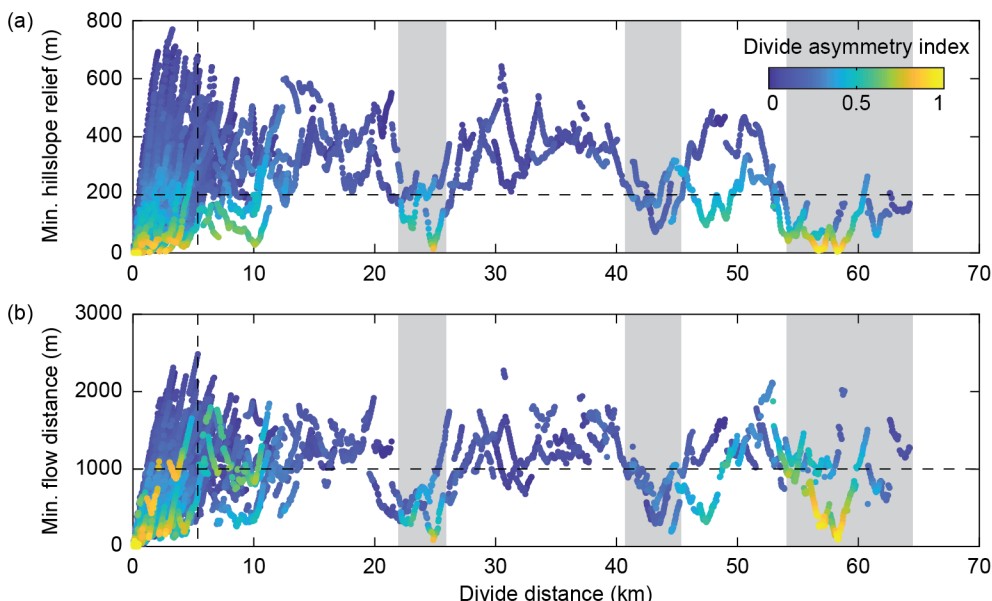

**Figure 11:** Minimum hillslope relief (a) and minimum flow distance (b) along the divide network of the Big Tujunga catchment. Colours denote the divide asymmetry index (*DAI*). Black stippled lines indicate thresholds used to identify anomalous divides in Figure 12. Grey shaded areas highlight regions with anomalously low hillslope relief and flow distance.

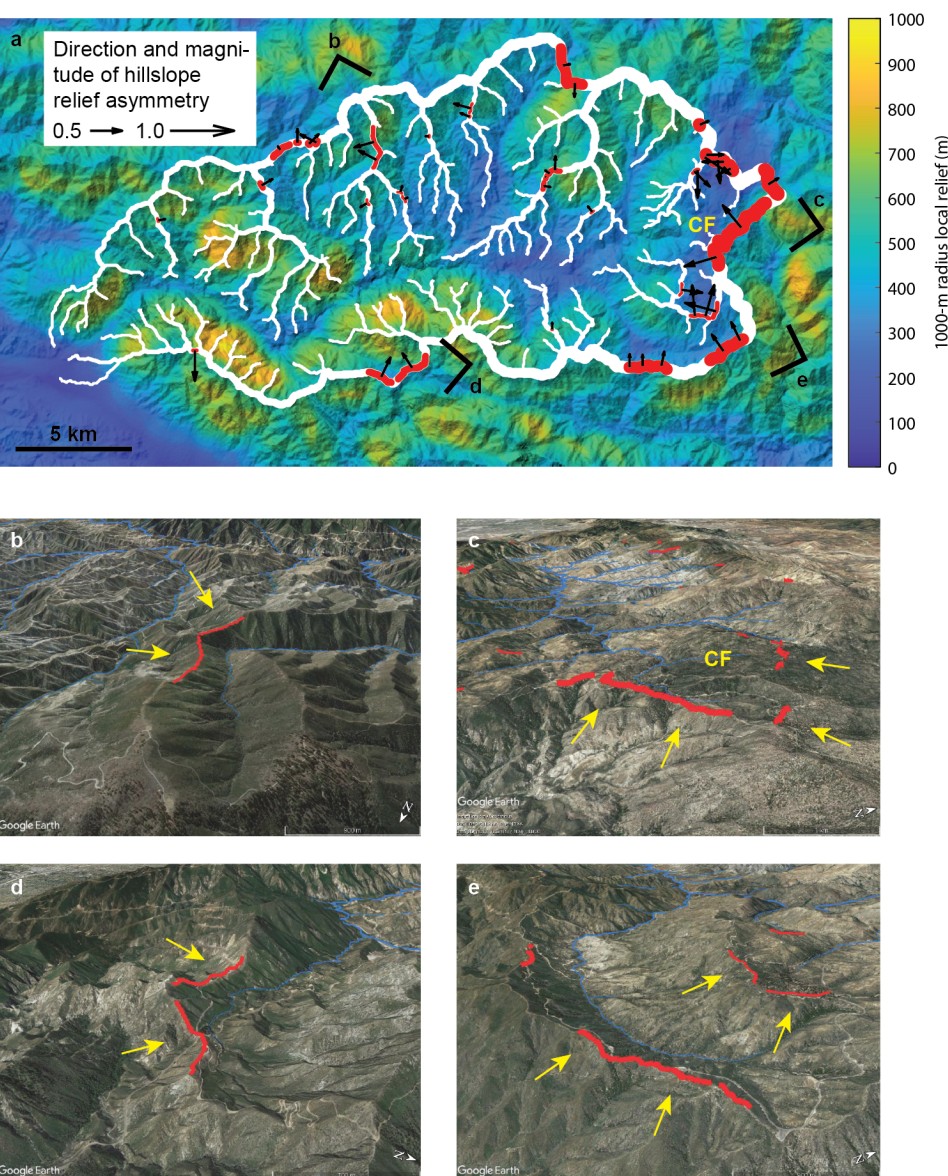

**Figure 12:** Anomalous divides in the Big Tujunga catchment. (a) 1000-m radius local relief map draped over hillshade image. White lines show the divide network and red lines depict asymmetric (*DAI*>0.4) divide edges with minimum hillslope relief <200m and minimum flow distance <1000m. Black arrows indicate the direction and magnitude the *DAI*, with the arrow pointing in the direction of lower relief, i.e., the inferred direction of divide migration. (b-e) Oblique Google Earth views of asymmetric divides shown in (a). CF = Chilao Flats.

# Drainage divide networks, Part 2: Response to perturbations

Dirk Scherler[1,2], Wolfgang Schwanghart[3]

[1] GFZ German Research Centre for Geosciences, Section 3.3, Telegrafenberg, D-14473 Potsdam, Germany
[2] Institute of Geological Sciences, Freie Universität Berlin, D-14195 Berlin, Germany
[3] Institute of Environmental Sciences and Geography, University of Potsdam, D-14473 Potsdam, Germany

*Correspondence to*: Dirk Scherler (scherler@gfz-potsdam.de)

**Abstract.**

Drainage divides are organized into tree-like networks that may record information about drainage divide mobility. However, views diverge about how to best assess divide mobility. Here, we apply a new approach of automatically extracting and 10 ordering drainage divide networks from digital elevation models to results from landscape evolution model experiments. We compared landscapes perturbed by strike-slip faulting and spatiotemporal variations in erodibility to a reference model to assess which topographic metrics (hillslope relief, flow distance, and $\chi$) are diagnostic of divide mobility. Results show that divide segments that are a minimum distance of ~5 km from river confluences strive to attain constant values of hillslope relief as well as flow distance to the nearest stream. Disruptions of such pattern can be related to mobile divides that are lower than 15 stable divides, closer to streams, and often asymmetric in shape. In general, we observe that drainage divides high up in the network, i.e., at great distances from river confluences, are more susceptible to disruptions than divides closer to these confluences and are thus more likely to record disturbance for a longer time period. We found that across-divide differences in hillslope relief proved more useful for assessing divide migration than other tested metrics. However, even stable drainage divide networks exhibit across-divide differences of any of the studied topographic metrics. Finally, we propose a new metric 20 to quantify the connectivity of divide junctions.

# 1    Introduction

Divide migration is a time-dependent process that is difficult to quantify. While the effects of regional-scale drainage captures

may be preserved within sedimentary archives (e.g., Clift et al., 2006), this is unlikely for smaller scale drainage captures or gradual divide migration. In such cases, most studies rely on topographic indicators. Mobile divides are typically inferred from post-drainage capture evidence: distorted drainage structures, low divides (wind gaps), or high tributary junction angles (e.g., Clark et al., 2004) (Figure 1). However, divide mobility may also be expressed in the topography without major drainage captures or flow reversals, but as a result from more gradual migration of divides. Willett et al. (2014) inferred drainage divide

mobility from across-divide differences of $\chi$ values, a proxy for steady-state river channel elevation (Perron and Royden, 2013). They argued that changes in drainage area within mountain ranges, e.g., due to tectonic strain of the crust (Yang et al., 2015), may commonly lead to relative differences in incision rate and the formation of low-relief landscapes that are bordered by migrating divides. Whipple et al. (2017a,b) however argued that the time scale of such changes is too short to profoundly affect mountainous landscapes. Instead, they argued that transient low-relief landscapes, such as those in Southeastern Tibet, are

more likely to be formed by regional changes in rock uplift rate and upstream propagation of knickpoints between the adjusted and unadjusted parts of landscapes. They also cast doubt on the ease of comparing across-divide differences in drainage network geometry (i.e., $\chi$ values) where the common base level is far and where opposing rivers may incise into areas of different rock types, different rock uplift rates, or different climates. Whipple et al. (2017a,b) instead proposed that the shape of drainage divides themselves holds clues about their mobility (Figure 1). Amongst the topographic parameters that they

tested are across-divide differences in channel elevation at a reference drainage area, mean headwater hillslope gradient, and mean headwater local relief.

In summary, several different metrics have been proposed that may allow quantification of divide mobility in both natural and modelled landscapes. Forte et al. (2018) compared the performance of these metrics with a landscape evolution model in which they induced divide mobility and concluded that across-divide differences in relief or gradient better depict divide motion than

$\chi$. In their analysis however, they focused on divide motion that is perpendicular to the regional drainage direction and averaged divide migration rates as well as topographic metrics across the entire width of the model domain, so that each time step is associated with single values for divide migration rate, erosion rate difference, and the tested topographic metrics. In part 1 of this study (Scherler and Schwanghart, 2020a), we presented a new approach for the automatic identification and ordering of drainage divide networks in a DEM, which removes the necessity to manually select drainage divides for comparison. Here,

we present experiments with a numerical landscape evolution model that we conducted to investigate how drainage divide networks respond to different perturbations, including fault activity and differences in erodibility. In contrast to previous studies that examined the response of drainage divides to perturbations, we studied the entire drainage divide network in an objective manner and examined how different portions of the divide network respond to perturbations. In addition, we tested the utility of a new metric that quantifies the connectivity of drainage divide junctions.

## 2    Materials and Methods

### 2.1    Landscape evolution model

We studied the response of divide networks to stream captures and divide migration, using the TopoToolbox Landscape Evolution Model (TTLEM; Campforts et al., 2017). In our experiments, we modelled the topographic evolution of a 20 km × 20 km square block (50-m node spacing), subject to uniform rock uplift, stream power-based fluvial incision (e.g., Howard and Kerby, 1983; Whipple and Tucker, 1999), and hillslope diffusion (e.g., Culling, 1963):

$$\frac{dz}{dt} = U - K_r A^m S^n - \nabla \boldsymbol{q_s} \qquad \text{Eq. 1,}$$

where $z$ is elevation (L), $t$ is time (T), $U$ is rock uplift rate (L T$^{-1}$), $A$ is upstream area (L$^2$), $S$ is local channel slope (L L$^{-1}$), $K_r$ is a parameter of the efficiency of river incision (T$^{-1}$) ($K_r = 1 \times 10^{-5}$ yr$^{-1}$), and $m$ and $n$ are dimensionless constants with values of 0.5 and 1, respectively. The last term on the right-hand side depicts elevation change due to the divergence in diffusive hillslope transport $\boldsymbol{q_s}$ (L$^3$ L$^{-1}$ T$^{-1}$), which we consider to be a linear function of hillslope gradient: $\boldsymbol{q_s} = -D\nabla z$, where $D$ is the diffusivity (L$^2$ T$^{-1}$) of soil creep ($D = 2 \times 10^{-3}$ m$^2$ yr$^{-1}$). All four edges of the block were fixed in elevation ($z = 0$ m), which forced rivers to flow outwards. The uplift rate ($U = 1$ mm yr$^{-1}$) was constant in all models. Our choice of parameter values was guided by the study of Whipple et al. (2017b), who tested a wide range of rock uplift and erosional efficiency parameters and found almost no difference of divide mobility in models with and without hillslope diffusion and for $n$ values of 1 and 2.

We started from a flat surface with imposed random noise and ran the experiment for 30 Myr, until the topography reached a steady state. The result of this model, which we termed 'Initialize', provided the starting point for four other models that we ran for 10 Myr (Figure 2). The model 'Reference' included no further changes. In the model 'Rotating', we included a circular (10-km diameter) left-lateral strike slip fault that was active throughout the experiment. Strike-slip faults are well known for enforcing drainage captures and thus divide mobility (e.g., Castelltort et al., 2012; Duval and Tucker, 2015). Although the rotating block has, to our knowledge, no real-world equivalent, this model setup represents a convenient way of simulating extended periods of strike-slip faulting, as the fault does not intersect the model boundary (Braun and Sambridge, 1997). The fault slip rate was fixed at 4 mm/yr, which corresponds to an angular velocity of $8 \times 10^{-7}$ rad/yr, resulting in ~460° of total rotation during the model run. We note that the rotating movement requires interpolation and thus leads to numerical diffusion of elevations within the rotating disc. However, the resulting change in total volume by interpolation is <0.03% of the volume uplifted during the same time and therefore small. The model 'Inclined' included 1-km thick and 5-km spaced layers of 50% reduced erosional efficiency of rivers ($K_r$), dipping 30° towards northwest. The 'Inclined' model is representative of a landscape in which rivers incise into tilted sedimentary rocks of non-uniform rock strength, similar to what has been studied by Forte and Whipple (2018). During the experiment, the combination of surface lowering and inclination resulted in the resistant layers regularly sweeping from southeast to northwest across the simulated landscape. The model 'Spheres' included 30 randomly assembled spheres of 3 km diameter with 75% reduced erosional efficiency of rivers ($K_r$). This experiment may

represent incision of a region that is characterized by country rocks with more-resistant magmatic intrusions. The expected

behaviour of this model is similar to the landscape response to localized perturbations studied by O'Hara et al. (2018).

## 2.2 Topographic analysis

We analysed the modelled topography and the associated drainage divide network. For each modelled topography and at each

time step (dt = 40,000 years), we first computed flow directions and flow accumulation, and subsequently identified the stream

network using a drainage area threshold of 0.2 km$^2$. We next derived the drainage divide network on the basis of the stream

network and using the algorithm proposed in Scherler and Schwanghart (2020a). We calculated divide distances as well as

divide orders based on the *Topo* ordering scheme (Scherler and Schwanghart, 2020a). As topographic metrics, we included

elevation ($z$), as well as hillslope relief ($HR$) and horizontal flow distance to the stream network ($FD$). $HR$ was measured as

the elevation difference between a point on the divide and the nearest river location as measured by the distance along local

flow directions. We also computed $\chi$ values on either side of a divide, using a reference area $A_0$ of 1 m$^2$, a reference concavity

$\theta_{ref}$ of 0.45, and setting the base level $x_b$ to 0 at the edge of the model domain (e.g., Perron and Royden, 2013):

$$\chi = \int_{x_b}^{x} \left( \frac{A_0}{A(x')} \right)^{\theta_{ref}} dx' \qquad \text{Eq. 2.}$$

For each divide edge, we computed these topographic metrics as well as the erosion rate (Eq. 1) for the two neighbouring

pixels that belong to adjacent drainage basins and denoted the across-divide minimum, maximum, sum, difference, and average

in any one metric $X$, as $X_{min}$, $X_{max}$, $\sum X$, $\Delta X$, and $\bar{X}$, respectively. Erosion rates were based on the erosion rate of the first

downslope stream pixel to reduce the impact of local noise along hillslopes. Topographic metrics of entire divide segments are

based on those of the divide edges that it is composed of. For quantifying across-divide differences in topographic metrics as

well as erosion rates, irrespective of the actual values, we used normalized indices of the form $\Delta X / \sum X$. One such index that

we frequently used in our study is the divide asymmetry index ($DAI = |\Delta HR / \Sigma HR|$), which is the absolute value of the

normalized hillslope relief difference, and which ranges between 0 (symmetric) and 1 (most asymmetric).

The above described across-divide differences in topographic metrics essentially aim to quantify divide mobility. In contrast,

Spotila (2012) studied the stability of divides and argued that divide junctions and pyramidal peaks are more stable than solitary

linear divides and might therefore act as anchor points for drainage divide networks. He proposed that divide junctions are

more difficult to erode than linear divides, due to their greater volume of topography per unit area, their greater mechanical

stability, and their reduction of confluent flows (Spotila, 2012). He also suggested that the stability of divide junctions is related

to the number of joining drainage divides. Because divide junctions obtained from our algorithm cannot connect more than

four divides segments (Scherler and Schwanghart, 2020a) – and most often connect three segments – we introduce a new

metric to quantify divide junction connectivity, $C_J$:

$$C_J = \frac{dx}{d_{d,max}} \sum_{i=1}^{n} \frac{d_i}{d_{d,i}} \qquad \text{Eq. 3.}$$

We define $C_J$ to correspond to the sum of the ratios of the Euclidean distance, $d$, and the divide distance, $d_d$, of all divide edges, $n$, within a specified maximum divide distance, $d_{d,max}$, times the ratio of the cell size, $dx$, and $d_{d,max}$. The dimensionless quantity $C_J$ is sensitive to the number of divides within a given divide distance from a junction, weighted by their orientation towards the junction (Figure 3). The value $d_{d,max}$ reflects the divide distance over which differences in junction connectivity are measured. For junctions that connect a constant number of straight and infinitely long divide segments, $C_J$ is not sensitive to the value of $d_{d,max}$. However, for actual junctions, $C_J$ is typically sensitive to the value of $d_{d,max}$, because as $d_{d,max}$ grows, increasingly more junctions are at a distance $d_d < d_{d,max}$ of a specific junction and thus the number of divide segments grows with $d_d$ (Figure 3a). In general, $C_J$ will be sensitive to the position of a junction within the drainage divide network, if the junction's maximum divide distance from an endpoint is smaller than $d_{d,max}$. In other cases, $C_J$ will provide a measure of how connected a junction is within a network, or, in other words, how prominent the junction is compared to other junctions in the network. In this study, we used a $d_{d,max}$ value of 5 km.

## 3 Results of numerical experiments

### 3.1 General behaviour

The simulated landscapes along with their drainage divide networks at the end of the numerical experiments are shown in Figure 4 and Figure 5, and in the Supplementary Material (Scherler and Schwanghart, 2020b) we provide movies of all simulations. To provide a measure of the mobility of drainage divides, we computed the percentages of drainage area that were exchanged during the simulations between individual catchments that drain to the margin of the model domain (Figure 6). Except for the Reference model, all models are characterized by notable changes in drainage area and mobile drainage divides. Area changes in the Initialize model are large in the beginning but level off rapidly during the first 1 Myr. Although area changes are small after 1 Myr, they continue for another 20 Myr, during which they are mostly decreasing. In the Rotating model, large area changes appear as discrete pulses, induced by drainage captures of major streams (Figure 5b), whereas the background area changes during the rotation/faulting are relatively small (<0.1%/40kyr). Area changes in the Inclined model are moderate (~0.25%/40kyr) throughout the simulation and oscillate in conjunction with the passage of more resistant layers through the landscape (Figure 5c). Area changes in the Spheres model are generally more pronounced if the resistant spheres appear in the course of rivers, which forces them to steepen and to induce surface uplift upstream, as opposed to their appearance at drainage divides, which increases the height of the divide, but which does not induce drainage divide migration at larger scale (Figure 5d).

### 3.2 Network topology

We first analysed the response of the entire drainage network topology to the perturbations by quantifying the aggregated length of divide segments as a function of their order (Figure 7). The first few Myr of the Initialize model are characterized by large changes in divide lengths and orders. Initially, the divide network extends to orders as high as 100, but rapidly contracts

as the drainage network becomes dendritic. After about 5 Myr, the highest orders are down to 60. Subsequent changes result in some scatter of the divide lengths, but not in the range of divide orders. Compared to the Reference model, in which the divide network structure does not change anymore, the Rotating, Inclined, and Spheres models exhibit changes in the divide network, mostly at divide orders greater than ~20. This observation is related to the fact that low-order divides are distributed across the entire model domain and their number is accordingly high. Any of the perturbations we imposed only affects some of these divides, and thus the impact on their average length is rather small. In contrast, high-order divides are constrained to the highest parts of the modelled land surface and their numbers are much lower. The imposed perturbations typically affect a greater portion of them and hence the scatter in divide lengths is wider. In the Rotating and Spheres models, we also observed that maximum divide orders occasionally extend to higher values, but these changes are rather small. We note that the above observations also prevail when considering divide distance instead of divide order, because the two are linearly related (Scherler and Schwanghart, submitted), with divide distance ~430 m × divide order. The 430 m corresponds to the mean length of the divide segments.

### 3.3 Topographic metrics

We next studied how the above-described disturbances affect drainage-divide metrics during the simulations (Figure 8). For all models, we computed the averages of topographic parameters measured at drainage divides of specific divide distance intervals (Figure 8a-d). As in the analysis of divide segment lengths by order, it should be kept in mind that the numbers of divide segments, or their aggregated lengths (Figure 7), are much higher for low orders and distances as compared to higher ones. For reference, a divide order of 20 corresponds to a divide distance of approximately 9 km. In the Initialize model, all of the studied metrics attain a constant value that remains unchanged in the Reference model (Figure 8), and which may or may not depend on the divide distance. For example, the mean elevation and junction connectivity ($C_J$) clearly increase with divide distance, whereas the flow distance exhibits only minor dependence on divide distance, and hillslope relief appears unrelated to divide distance. The dependency of some metrics on divide distance is partly explained by the model setup. Although divides with low distances occur also at higher elevation, the bulk of them are near the model edge, close to zero elevation. In contrast, divides at high distances are exclusively found near the centre of the model, where elevations are also high. Similarly, the junction connectivity ($C_J$) is high in the model centre, where the divides are far from most of the endpoints, which are more abundant near the edges of the model.

It is also worth noting that none of the normalized across-divide differences in the topographic metrics attain zero values in the Reference model. This means that even at topographic steady state, there exist residual across-divide differences in hillslope relief, flow distance and $\chi$. In the case of $\chi$, these also depend on the divide distance, and are greater closer to the model edge, where divide distances are low. In the perturbed models, we observed fluctuations in all topographic metrics, although of different magnitudes. For example, comparison of across-divide differences in erosion rate with differences in hillslope relief, flow distance and $\chi$ (Figure 8e-h) shows that the normalized difference in hillslope relief (i.e., the divide asymmetry index) is sensitive to drainage divide mobility in all perturbed models, whereas across-divide differences in $\chi$ and flow distance are

sensitive to divide mobility in the Rotating model, but less so in the Inclined and Spheres models. The junction connectivity ($C_J$) metric attains temporally-averaged values in the perturbed models that are quite similar to the constant values in the Reference model. In many cases, the deviations from the Reference model are greater the higher up in the divide network, i.e., for higher divide distances. This pattern is particularly well visible in the Inclined model, where the amplitudes of the oscillations in all of the parameters increase with divide distance.

### 3.4 Minimum hillslope relief and flow distance

Motivated by the observation of constant values in hillslope relief and flow distance in the Reference model, as well as in actual landscapes (Scherler and Schwanghart, 2020a), and by our expectation that small values in either one would be found where one catchment loses area to another (Figure 1), we next compared how minimum hillslope relief ($HR_{min}$), minimum flow distance ($FD_{min}$), and the divide asymmetry index ($DAI$) vary with divide distance in the Reference model and the three models with landscape perturbances (Figure 9). In contrast to the average values in Figure 8, we provide these metrics for all divide edges during the last 1 Myr of the model runs. Note also that we plotted data points in an order which brings high $DAI$ values to the front, to better assess where asymmetric divides are located, but that in all four models, relatively high $DAI$ points may plot on top of low $DAI$ points. In the Reference model, both minimum hillslope relief and minimum flow distance reach relatively steady values ($HR_{min}$ ~250-350m; $FD_{min}$ ~400-600m) at a divide distance of ~5 km. At lower divide distances, both $HR_{min}$ and $FD_{min}$ approach zero – simply because divides are defined to start at the stream network – and these divides can become increasingly more asymmetric. It is notable, however, that some of the highest $HR_{min}$ and $FD_{min}$ values are also observed at low divide distances of approximately 1-2 km. In the three other models, the transition between quite variable divides at short distances and more steady 'background' values at higher distances appears to be preserved, but we observe generally more variability. For example, in the Rotating model, we observe divides with significantly lower $HR_{min}$ and $FD_{min}$ values at higher distances. These divides are particularly prominent at a distance of ~10 km and ~15-22 km and correspond to the position of the strike-slip fault. Where $HR_{min}$ and $FD_{min}$ are low, $DAI$ values are relatively high (divides are highly asymmetric), although there also exist divides that have high $DAI$ but regular $HR_{min}$ and $FD_{min}$ values. In the Inclined and Spheres models, the $HR_{min}$ and $FD_{min}$ values are never as low as in the Rotating model at divide distances >5 km, which reflects the lack of drainage captures. Instead, we observe frequent excursions to both higher and lower $HR_{min}$ and $FD_{min}$ values, either across all divide distances (Inclined) or at specific locations (Spheres). Deviations from the average values in the Reference model are greatest in the Spheres model compared to all other perturbed models and always correspond to peaks that grow where strong spheres are exhumed. In the three disturbance models, $DAI$ values are generally higher than in the Reference model – although divides with high $HR_{min}$ and $FD_{min}$ values and at great divide distances mostly appear to have somewhat lower $DAI$ values.

## 3.5 Junction connectivity

The spatial pattern of divide junction connectivity ($C_J$) values at the end of the landscape evolution experiments (Figure 10) partly follows the pattern of divide distances (Figure 4). Junctions with higher $C_J$ values tend to occur at higher elevation and at greater divide distance ($d_d$). In the Reference model, the highest $C_J$ values occupy the centre of the model domain as well as the centres of the four quadrants of the model domain, resembling the five of a six-sided dice. In the Rotating model, these clusters of high $C_J$ values are maintained, but their connection is disrupted by the strike-slip fault, which induces low $C_J$ values. The centrally located divide junctions occupy a similar range in $C_J$ values, but are all shifted to higher elevations (Figure 4). Similar, although lower, offsets to higher elevations occur in the Inclined and Spheres models, where junctions coincide with rocks of reduced erodibility. In the Spheres model, the basic structure of $C_J$ values is similar to that of the Reference model, but the highest $C_J$ values are steered towards the less erodible spheres, where they also attain $C_J$ values that are distinctly higher than in any of the other models (Figure 10b). In general, divide junctions with combinations of elevation and $C_J$ values that are outside the range of values observed in the Reference model are found in the most disturbed parts of the landscape (Figure 10c). In summary, the perturbed models appear to induce mostly changes in junction elevation, whereas changes in junction connectivity ($C_J$) are seemingly constrained to the Spheres model.

## 4 Discussion

### 4.1 Quantifying drainage divide mobility

The analysis of stream networks has become a standard tool for inferring tectonic forcing and landscape history (e.g., Wobus et al., 2006; Kirby and Whipple, 2012; Demoulin, 2012; Schwanghart and Scherler, 2014). The divide network holds the potential for recording similar tectonic forcing, but also other aspects of landscape history (e.g., Willett et al., 2014). The question is, which divide metrics are useful to analyse and what they tell us about landscape history? Our Rotating model induced relatively sudden drainage captures (Figure 6). Because such events are associated with the dissection of drainage divides, reliable indicators are values of hillslope relief (*HR*) and flow distance (*FD*) that are much lower compared to the values that divides (> ~5 km divide distance) strive for (Figure 9). More gradual divide migration, however, likely lacks such simple diagnostic criteria, and in those cases, across-divide differences in topographic metrics may be more suitable indicators of divide mobility. The most commonly used metric to infer drainage divide mobility is the across-divide difference in $\chi$ (Willett et al., 2014). Although the utility of this metric has recently received some critique (Whipple et al., 2017b; Forte and Whipple, 2018), it has become a popular tool for studying drainage divides. Whipple et al. (2017b) and Forte and Whipple (2018) instead advocated the use of other topographic metrics, including mean gradient, mean local relief, and channel bed elevation, measured at or upstream of a reference drainage area. We note that these latter metrics are typically highly correlated and very similar to the hillslope relief and *DAI* metrics that we included in this study.

Figure 11 shows how normalized across-divide differences in $\chi$, hillslope relief (*HR*), and flow distance (*FD*) compare to normalized across-divide differences in erosion rate (*ER*), evaluated for each divide edge from the last 1 Myr of the landscape evolution experiments Reference, Rotating, Inclined, and Spheres. We find that across-divide differences in hillslope relief (*HR*; Figure 11a) are most sensitive to the disturbances included in the models, whereas across-divide differences in $\chi$ are similarly sensitive to disturbances in the Rotating model, but less so in the Inclined and Spheres models (Figure 11b; Figure 8). Across-divide differences in flow distance (*FD*) are the least sensitive to disturbances in the models and show the largest scatter when compared with erosion rates (Figure 11c). However, there also exists substantial scatter in the relationship between across-divide differences in hillslope relief and erosion rate, which partly depends on the divide distance. In general, we observe that the scatter is higher for divide distances <~5 km (dark blue in Figure 11), which corresponds to the value below which we observe large variability in divide morphology, even in the Reference model (Figure 9). To quantify the correlation of the normalized across-divide differences in topographic metrics with normalized across-divide differences in erosion rate, we fitted a linear model to all drainage divide edges from the entire model runs, categorized into 1-km divide distance bins, and show the resulting coefficients of determination ($R^2$) in Figure 12. As already suspected from Figure 8 and Figure 11, the $R^2$ values differ in between models and metrics, and also depend on divide distance. In general, we observe that all metrics perform poorly at divide distances < ~5 km, and that across-divide differences in flow distance perform poorly even at higher distances. The highest $R^2$ values are linked to across-divide differences in hillslope relief, whereas across-divide differences in $\chi$ attain similar $R^2$ values in the Rotating model, but some of the lowest $R^2$ values of all metrics in the Inclined and Spheres model. This difference may be explained by the fact that in the latter two models, we introduced spatial variability in the erosional efficiency of rivers ($K_r$) that we did not account for in our across-divide comparison of $\chi$, as would be required (Willett et al., 2014). In natural landscapes, however, these values and their variability are rarely well known.

We speculate that the influence of divide distance on topographic metric-erosion rate relationships may also account for the differences in scatter observed by Sassolas-Serrayet et al. (2019) in landscape evolution experiments similar to our Initialize model between larger and smaller basin areas. But even when excluding divides of low order or low divide distance, we still observe considerable scatter in the topographic metric-erosion rate relationships, which, at the very least, demands caution when interpreting divide morphology in terms of mobility. In this regard, Figure 5 and studying the videos of the landscape evolution experiments (see Supplementary Material) is insightful: where drainage divides are migrating, one typically observes a range of across-divide topographic metric values that vary considerably during the migration. In other words, despite a continuous divide migration at large scale, there often exists small-scale variability in divide morphology that may in part be related to across-divide differences in topographic metrics lagging behind across-divide differences in erosion rate.

As a final note, we emphasize that the above observations are from our numerical experiments, which depict an idealized world. It is clear that complexities present in nature, such as anisotropic and variable rock properties, hydro-climatic gradients, mass wasting events or biological influences on erosion processes and rates can lead to landscape patterns which bias any of the above topographic metrics, and need to be taken into account when inferring divide dynamics from divide metrics in natural landscapes.

## 4.2    Divide network dynamics

Stream networks tend to attain configurations that are in equilibrium with the geological and climatic environment, given an initial condition (e.g., Rinaldo et al., 2014). Because drainage divides are defined by adjacent drainage basins, the geometry of divide networks should attain a similar equilibrium, which expresses itself both in the geometry of divides and in the topology of divide networks. Our numerical experiments have shown that during the initial establishment of a stream network, on a relatively flat surface, both stream and divide networks are far from their steady-state configuration and characterized by networks that extend to high orders (Figure 7), and long divide distances. During the subsequent extension and shrinkage of individual streams towards their steady-state configuration, the divide network contracts and primarily high-order divide segments shorten and become fewer, whereas divides of low orders maintain their frequencies (Figure 7).

In general, divide segments of high order, i.e., at great distance from endpoints, appear to be the most responsive to landscape disturbances (Figure 8). In the case of the Rotating model, this is in part expected, because the inner rotating part of the landscape contains the highest order divide segments (Figure 4b). In the cases of the Inclined and Spheres models, it may be related to the increased probability of recording a disturbance, because the adjoining basins cover a larger area compared to lower-order divides. In other words, if drainage captures happen somewhere within a drainage basin, this will most likely influence divides further upstream. Over a distance of less than ~5 km from divide network endpoints, the divide segments transition from low interfluves at river junctions to high topographic ridges, as seen in the Reference model (Figure 9). In the other models, most of the investigated morphometric parameters are quite variable over the same distance and can be seen to rapidly adjust to disturbances such as drainage captures or migrating divides. Such behaviour is consistent with the observation that the timescale of a rivers' response to changes in drainage area increases with the distance from the divide to the outlet of a river (Whipple et al., 2017b). To reliably distinguish the morphologic effects of real disturbances from 'noise' close to the river, a minimum divide distance of perhaps ~5 km, as in our analysis of the Big Tujunga divide network (Scherler and Schwanghart, 2020a), appears appropriate. This minimum divide distance could be lower or higher, depending on factors like drainage density and average hillslope relief, for example.

Our new junction connectivity index ($C_J$) complements existing topographic metrics in assessing divide network dynamics. For example, the junction connectivity in our Rotating model is low along the fault (Figure 10), consistent with the absence of stable divides. In the Spheres model, however, the appearance of more resistant rocks at the surface often resulted in migration of divides towards the spheres (Figure 5c, Supplementary Material; Scherler and Schwanghart, 2020b). In this case, parts of the drainage divide network were mobile, not stable, but they moved towards particularly stable portions in the landscape. Therefore, the junction connectivity index ($C_J$) may also be interpreted as an attractor or centrality index (Phillips et al., 2015) that quantifies how strong a drainage divide network has been pulled towards and anchored at a certain junction (Spotila, 2012).

## 5    Conclusions

Based on landscape evolution model experiments in which we forced divides to migrate, we found that stable drainage divides strive to attain a constant hillslope relief as well as flow distance from the nearest stream, provided a sufficiently large divide distance to avoid confounding influences near the edges of the divide network. In our experiments this distance is ~5 km from endpoints. Simple indicators of mobile divides are anomalously low hillslope relief or flow distance values, which could signal beheaded valleys or future capture events. Overall, drainage divides located high up in the network, i.e., at great distance from endpoints, are more vulnerable than divides closer to endpoints of the network and are more likely to record disturbance for a longer time period. In our comparison of different topographic metrics to assess drainage divide mobility, we found that across-divide differences in hillslope relief proved more useful for assessing divide migration than other tested metrics.

## 6    Code availability

The divide algorithm developed in Scherler and Schwanghart (2020a) has been implemented in the TopoToolbox v2 (Schwanghart and Scherler, 2014). The codes will be made available with the next TopoToolbox release, and shall be accessible through https://github.com/wschwanghart/topotoolbox.

## 7    Supplement

The supplement related to this article is available online at: http://doi.org/10.5880/GFZ.3.3.2019.005 (Scherler and Schwanghart, 2020b).

## 8    Competing interests

The authors declare that they have no conflict of interest.

## 9    Acknowledgements

D. Scherler acknowledges funding through grant SCHE 1676/4-1 from the Deutsche Forschungsgemeinschaft (DFG).

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

## 11   Tables

## 12   Figure captions

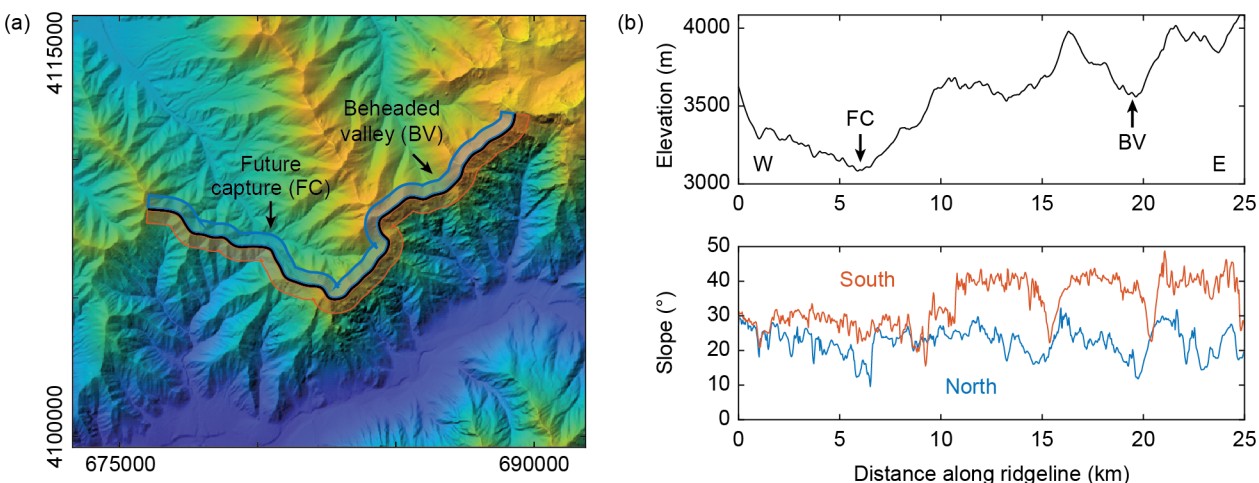

**Figure 1:** Example of mobile drainage divide in the Hindukush, Afghanistan. (a) Digital elevation model draped over hillshade image. Colours denote elevations that range from ~2000 m (blue) to ~4000 m (yellow). Black line traces drainage divide with clear signs of mobility. Centre of map lies at approximately 37.10°N and 71.06°E. North is up and the UTM zone is 43. (b) Profiles of elevation (upper panel) and
mean slope (lower panel) along the drainage divide shown in (a). Mean slope angles were computed along swath profiles that extend 500 m to the north (blue) and south (red) of the drainage divide.

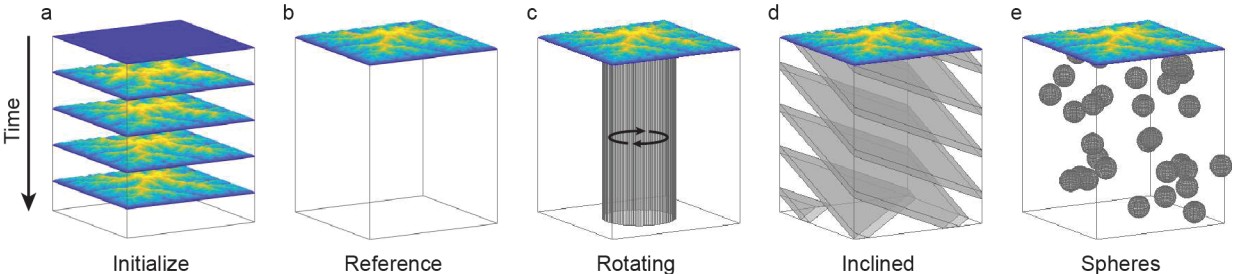

**Figure 2:** Graphical representation of the model setups in the landscape evolution experiments. The starting condition of each model is shown at the top of each rock column. During the experiments, the entire rock column is eroded and deeper lying regions are exhumed. (a) 'Initialize' started from a nearly flat surface and reached almost steady state topography; the end of 'Initialize' provided the starting point for all other models. (b) 'Reference' included no changes. (c) 'Rotating' included a circular left-lateral strike slip fault, active throughout the experiment. (d) 'Inclined' included 1-km thick and 5-km spaced layers of 50% reduced erodibility, dipping 30° towards northwest. (e) 'Spheres' included 30 randomly assembled spheres of 3 km diameter with 75% reduced erodibility.

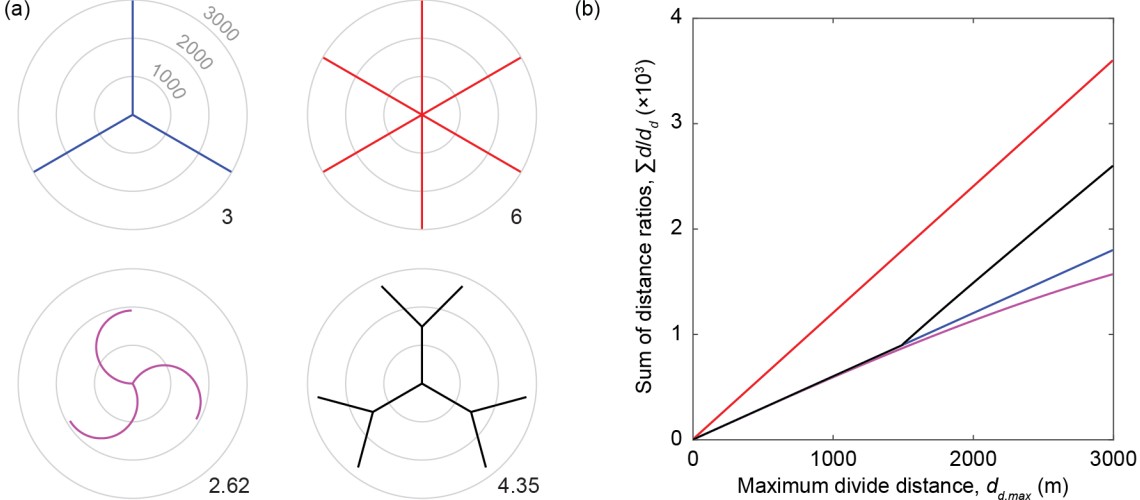

**Figure 3:** Illustration of junction connectivity ($C_J$) of different drainage divide junctions. (a) Map-view of four hypothetical divide junctions. Grey circles indicate Euclidean distance from the junction. Numbers to the southeast in each map gives the $C_J$ value, based on a $d_{d,max}$ of 3000 m. (b) Sum of distance ratios ($\sum \boldsymbol{d}/\boldsymbol{d_d}$; cf. Eq. 3) as a function of the maximum divide distance for each case in (a).

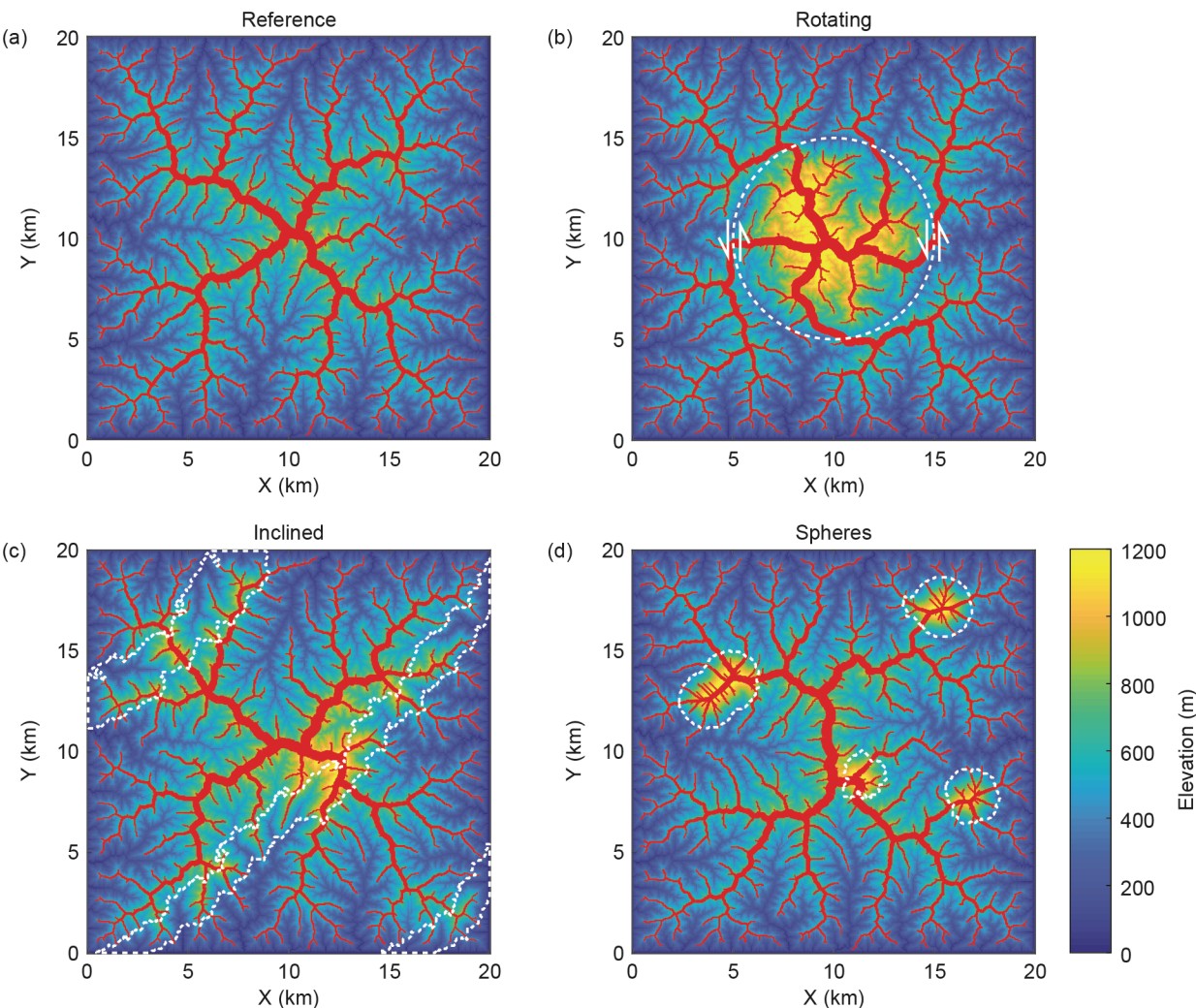

**Figure 4:** Modelled topography and drainage divide network (red solid lines) at the end of the landscape evolution experiments: (a) Reference, (b) Rotating, (c) Inclined, and (d) Spheres. White stippled lines show strike-slip fault in (b) and regions with reduced erodibility in (c) and (d).


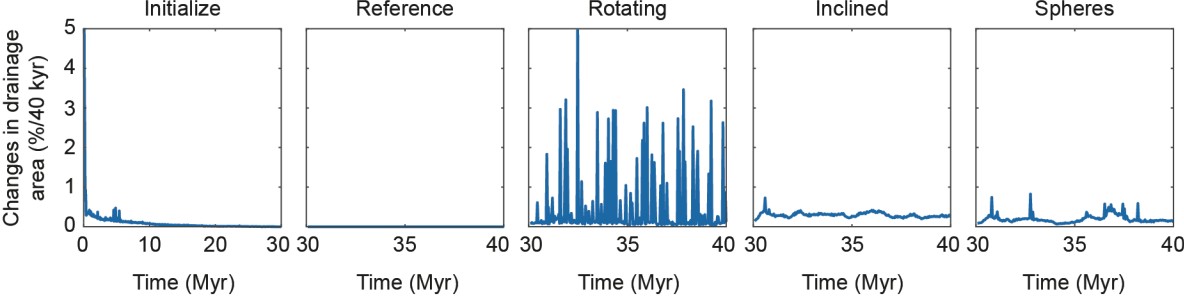

**Figure 5:** Erosion rates and divide asymmetry index (DAI) at the end of the landscape evolution experiments: (a) Reference, (b) Rotating, (c) Inclined, and (d) Spheres. Black stippled lines show strike-slip fault in (b) and regions with reduced erodibility in (c) and (d).


**Figure 6:** Changes in drainage area during the landscape evolution experiments. Y-axis unit is in percent per time step, which was 40 kyr.

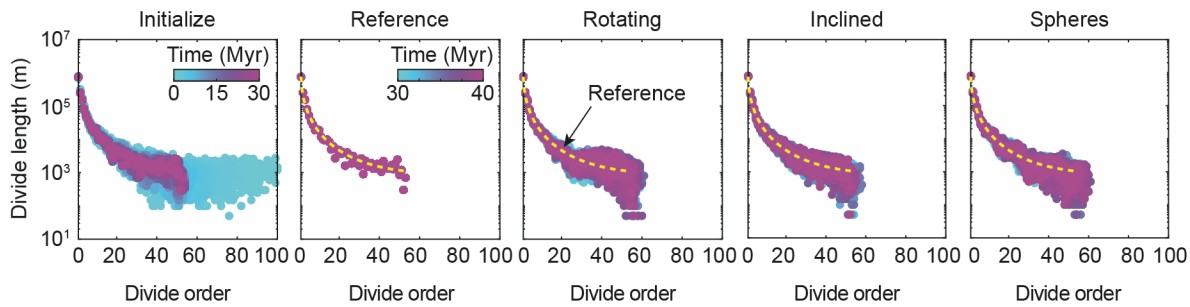

**Figure 7:** Changes in divide length for divides of different orders during the landscape evolution experiments. Divides have been ordered with the *Topo* ordering scheme.

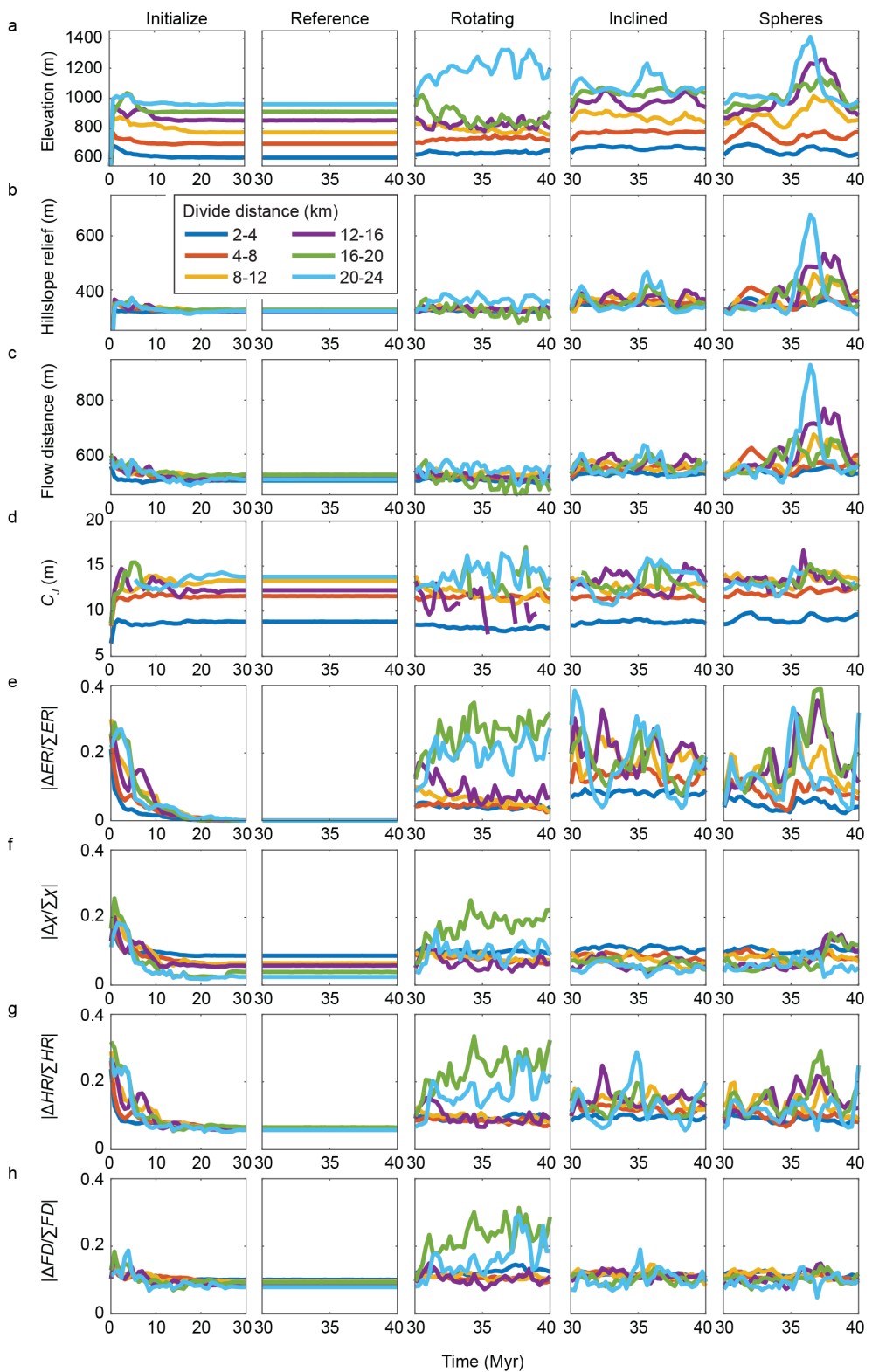

**Figure 8:** Temporal evolution of the drainage divide network during the landscape evolution experiments. Colored curves show mean values for divide segments at different divide distances. (a) Elevation. (b) Hillslope relief (*HR*). (c) Flow distance (*FD*). (d) Junction connectivity (*C_J*). (e) Across-divide difference in erosion rate (|*ΔER/ΣER*|). (f) Across-divide difference in chi (|*Δχ/Σχ*|). (g) Across-divide difference in hillslope relief (|*ΔHR/ΣHR*|); which is equal to the divide asymmetry index (*DAI*). (h) Across-divide difference in flow distance (|*ΔFD/ΣFD*|).

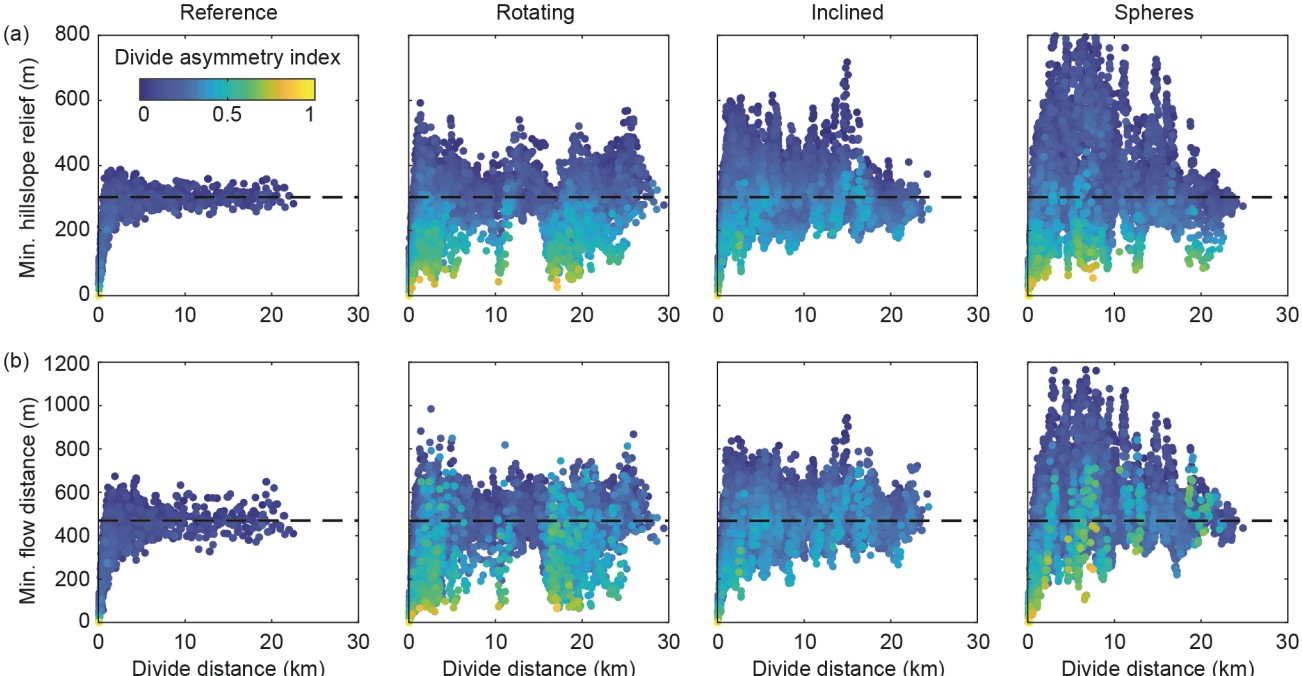

**Figure 9:** Minimum hillslope relief (a) and minimum flow distance (b) of all divide segments along the drainage divide network from the last 1 Myr of the landscape evolution experiments Reference, Rotating, Inclined, and Spheres. Colours denote the divide asymmetry index (*DAI*). Black stippled line indicates the average values of minimum hillslope relief and minimum flow distance from the model Reference, measured between a divide distance of 10 and 20 km.

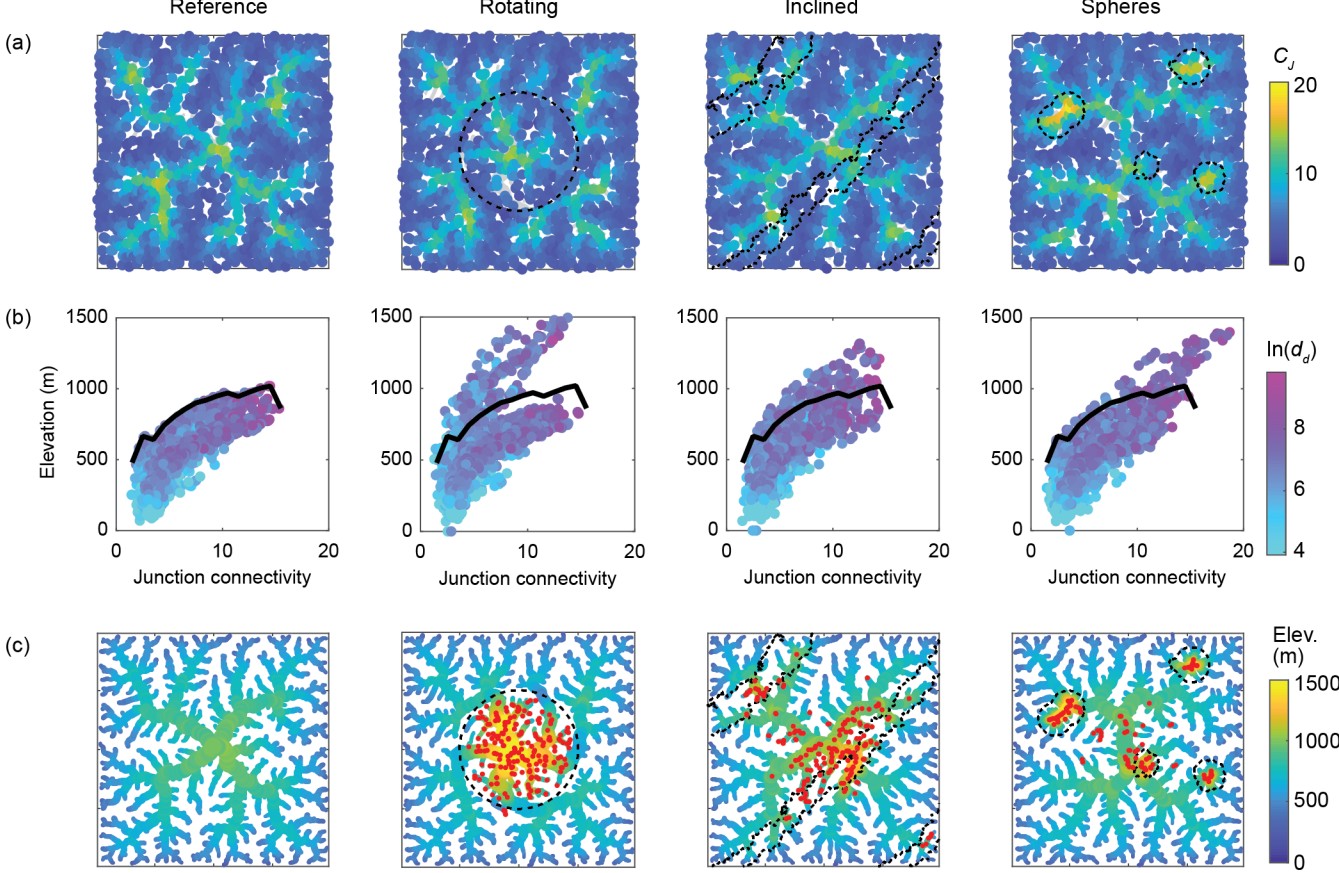

**Figure 10:** Divide junction connectivity ($C_J$) from the final stage of the landscape evolution experiments Reference, Rotating, Inclined, and Spheres. $C_J$ was calculated with a $d_{d,max}$ value of 5000 m. (a) Map-view of divide junction $C_J$ values. Black dashed lines indicate perturbation elements in the models. (b) Scatter plot of $C_J$ by elevation, coloured by the natural logarithm of the divide distance ($d_d$). Thick black line follows upper boundary of divide junctions in the Reference model. (c) Drainage divide network coloured by elevation. Red dots correspond to junctions in the perturbed models that lie higher than the junctions in the Reference models.



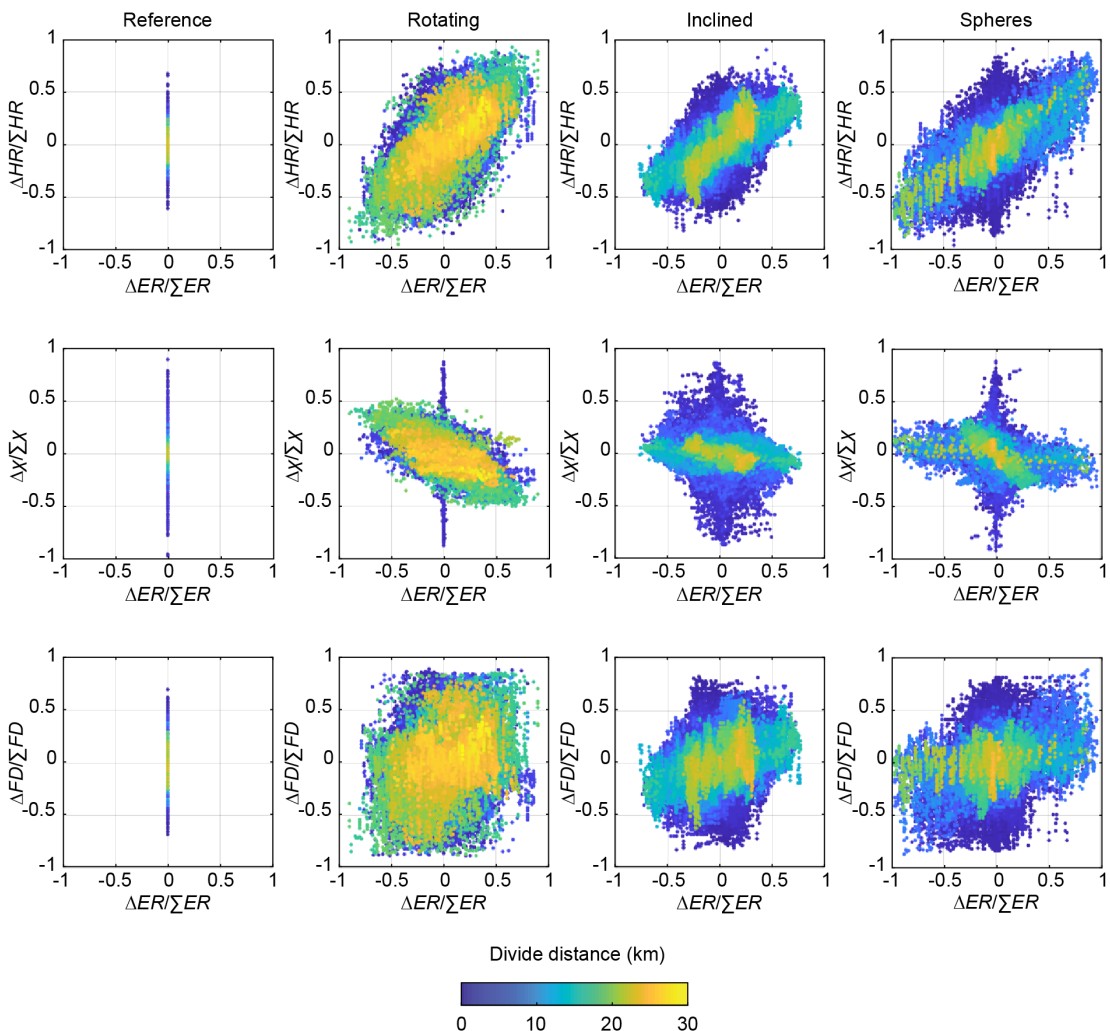

**Figure 11:** Relationship between across-divide differences in the topographic metrics hillslope relief (*HR*), chi (*χ*), and flow distance (*FD*), with across-divide differences in erosion rates (*ER*) for all divide edges in the last 1 Myr of the landscape evolution experiments Reference, Rotating, Inclined, and Spheres. Marker colour denotes the divide distance ($d_d$). Note that the markers are sorted by their divide distance to show the highest divide distance above data points with lower divide distance, and that data points with lower divide distance occur beneath those with higher divide distance.


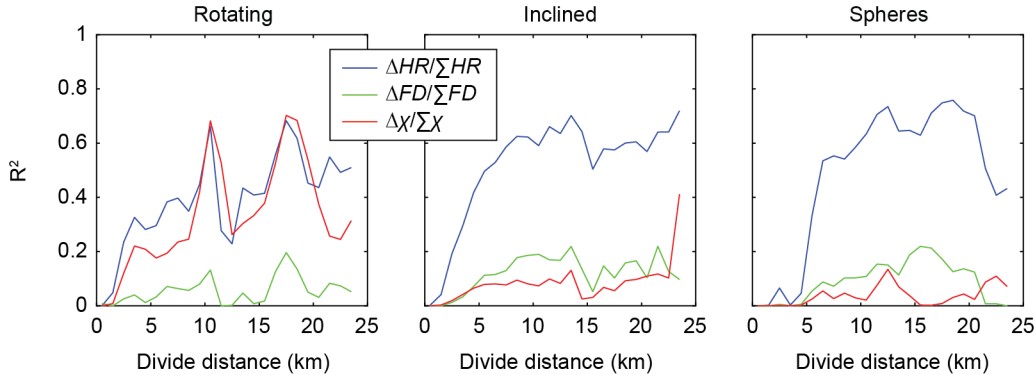

**Figure 12:** Coefficient of determination for linear regressions between normalized across-divide differences in different topographic metrics (*HR* = hillslope relief; *FD* = flow distance; *χ* = chi) and normalized across-divide differences in erosion rates, as a function of drainage divide distance in the three perturbation models.