# Peer review of "Drainage divide networks, Part 1: Identification and ordering in digital elevation models"

_Earth Surface Dynamics, 2019_

## Referee Comment (RC1) · Anonymous Referee #1 · 24 Oct 2019

Referee report on "Identification and ordering of drainage divides in digital elevation models" by Scherler & Schwanghart

The manuscript submitted by Scherler & Schwanghart proposes a new way to calculate and characterize drainage basin boundaries and shows how the new metrics can be used to identify sections of the divide, which are vulnerable to topographic changes.

The paper presents a valuable contribution to geomorphology especially as the tools are made freely available in a easily usable framework (topotoolbox), which is accessible also for IT-skilled geomorphologists.

While the technical part of the manuscript is well presented and the algorithms clearly

described, the analysis section lacks a central theme. Instead of getting to a point which shows how landscape perturbations affect the geometry or better topology of drainage divides the authors rush in a rather descriptive way through a series figures (Fig 11-21) most with multiple sub-pannels.

Given that the title focuses on "algorithm and metrics" with the analysis only being an "application showcase", I do not see this as a critical part for the paper. However the authors may consider shortening this part and focusing on only one perturbation scheme (maybe the rotating one which shows the most striking responses) which exemplifies their metrics without trying to make a geomorphological claim.

A more thorough analysis of different perturbations and their impact on drainage divide networks may be presented in a separate paper.

Below a few minor comments and questions/suggestions:

For the efficient calculation of watersheds, as well as their susceptibility to perturbations by topographic changes, I would like to draw the author's attention to a series of papers which may not be on the general "radar" of the geomorphology / hydrology audience:

Fehr, E., et al. "New efficient methods for calculating watersheds." Journal of Statistical Mechanics: Theory and Experiment 2009.09 (2009): P09007.

Impact of Perturbations on Watersheds E. Fehr, D. Kadau, J. S. Andrade, Jr., and H. J. Herrmann Phys. Rev. Lett. 106, 048501

Scaling relations for watersheds E. Fehr, D. Kadau, N. A. M. Araujo, J. S. Andrade, Jr., and H. J. Herrmann Phys. Rev. E 84, 036116

Usually, when making a topographic analysis stream networks are defined by a flow accumulation threshold. The authors define a parameter which describes the distance from the divide to the closest stream. Such a parameter would, however, depend on the specific choice of the flow accumulation threshold and could vary between 0 and the system size. The authors should add a formal definition of their stream networks

and clarify how they addressed this point.

The following comments/questions are using Figures as reference, but should also be addressed in the corresponding parts of the main manuscript.

Figure 1: Wouldn't it make sense to put (a) and (b) into a single (bigger) figure such that the reader directly can associate the divide network with the corresponding stream network?

Figure 3: Would it be possible to reduce the complexity of the figure without scarifying the main message. If I read the figure correctly, the right hand half contains already all necessary information.

Fig 7: The 3d figures are small hard to understand.

a. Initialize: I would suggest the authors to put an arrow "time" from the initial to the final (steady state condition). (this may also be presented simply in plain top view) c. Labeling the (rotating) circle and 12h initial position and showing how this marker moves through time (3 snapshots) may be easier to understand than the small 3d visualization d. and e. I had first difficulties to understand these graphics as they show the LEM surface and the "lithology" underneath, which changes the erodibility as the landscape erodes through this heterogeneous bottom. Here the authors may improve they description explicitly mentioning (depth) and the fact that the topography erodes through these formation creating a variable erodibility over time.

another option would be to combine c. d. e. as small panels in Fig. 11 and showing there 2 time snapshots in the evolution of the topography.

Fig 13: While the "Initialize" plot shows a significant change over the course of 10Myr, the changes in the plots of the perturbed systems seems almost insignificant, especially when compared to the changes of the "Reference" topography offer the course 10Myr-20Myr. Note, if I am correct, the "Reference" solution should have reached steady state and therefore should show no changes at all. In order to present how the

**ESurfD**
disturbance changes affect the divides it would make more sense to compare the disturbed system to the reference system at the same time, thus t=15Myr and 20Myr and show the difference between the two. Alternatively one could also evaluate the time evolution of the different perturbed cases relative to the evolution of the "Reference" solution.

Fig 17: Fig 17b consists of a main (invariant) cluster (black dashed line) corresponding to the general topographic shape of the landscape. It would be interested to see where (in the landscape) the deviations from this behavior are located.

Fig 21: The scatter plots in this figure are difficult to interpret, especially because color is an additional variable. e.g. do yellow points mask blue markers underneath? It may be favorable to divide the color scale in 3 or 4 categories and bin the results in x,y for each category. Additionally the authors should quantify between the different measured quantities using e.g. a simple linear regression model.

I hope that the above comments & suggestions help the author to improve the manuscript and strengthen their arguments.

---

## Referee Comment (RC2) · Anonymous Referee #2 · 29 Dec 2019

The authors present an algorithm to automatically extract the drainage divide networks from digital elevation models as well as ordering schemes to classify the different divides with respect to each other. The application of the algorithm to a field example (San Gabriel Mountains) and different numerical landscapes is shown, exploring the relationship between different geometric and topographic metrics attributed to divides with respect to their mobility.

I believe that the tool presented in this manuscript (and already implanted in Topotoolbox) is valuable to explore landscapes more systematically, offering the potential of developing new approaches to characterize landscape stability and to retrieve informa-

tion about the history and drivers of landscape evolution.

I also believe that Esurf is the perfect outlet for this contribution, but before I can recommend the manuscript for publication, I would like to point out some comments/concerns for the authors to consider in order to clarify certain parts of the manuscript, as well as to highlight the main points of this contribution.

**1 General Comment**

The authors have carried out an extensive analysis of divide drainage networks from a field case (Big Tujunga catchment) and several numerically simulated landscapes (including their temporal evolution) by exploring different (topologic, geometric and topographic) metrics. The manuscript contains 21 figures (with 120+ panels! – some of them quite complex). I think the number of figures is excessive, particularly because the presence of many of them (and their description) are not that insightful/necessary to make the main points on the paper, which are currently diluted. I would strongly suggest that the authors transfer to the Supplementary Material most of the panels, keeping only the ones that are key to the message, and expanding in their description and insight gained from this analysis. I believe the manuscript will substantially benefit from an effort in this direction, improving its readability and impact.

**2 Comments:**

- Line 76: It is stated that *divide networks even contain cycles*. Do cycles correspond to enclosed (internally drained) basins? In the rest of the manuscript, it is said that these networks are represented as tree-like networks, i.e. no cycles, (e.g. line 116) and that enclosed basins are not considered in the analysis. Is

**ESurfD**
the algorithm and ordering scheme applicable for the cases the network contains cycles?

- A new ordering scheme is introduced (Topo) and it is mostly adopted for the analysis reported in the manuscript. However, I missed a more detailed comparison between the different schemes in terms of characterizing the divide network.

- There are some metrics used or introduced in the manuscript, which although might seem intuitive, are not properly defined, and they can lead to misunderstandings. I would encourage the authors to properly define each of the metrics, particularly those that are central to the manuscript. As an example, I have struggled in imagining what is the definition of divide distance (which is used in almost every figure on the paper). The first time that it is mentioned in the manuscript is in line 129-131, where it is stated: *"We thus propose to order the nodes and edges of the divide network, by their maximum down-divide distance from divide endpoints, measured either in map units or in the number of divide segments. From now on, we call the distance measured 130 in map units along the divide network the divide distance (dd)."* What does *down-divide distance* mean? Only paths where elevation is strictly decreasing are considered? Please provide a clear definition of the different metrics utilized (consider including an illustration if needed – e.g. Fig 4 is quite helpful)

- Section 3, particularly section 3.1 and Appendix 7.1, are the central contribution in the paper. However, I find the description of the algorithm not very clear, particularly the second step (lines 140-143) and Figure 3, which I am unable to follow. There are terms that are not defined, e.g. how do you define (the number of) segment termini?

- Line 181 – Since the metrics are computed for each divide edge, isn't $\sum X$ just twice the value of $\bar{X}$? In that case, I guess there is no need to keep both.

- Equation 7. Not clear. Should $d$ and $d_d$ function of x within the integral?

- Line 240-241:*"In contrast, the ordering rules of the Strahler and Shreve scheme (see Eq. 1 and Eq. 2) may yield unequal orders during the sorting, so that the divide orders of the last divide segments may be different by more than one."* I can see how this can happen for the Shreve scheme but under what circumstances would that occur using the Strahler scheme?

- Line 254 – Why the analysis is limited to $\omega < 55$. Is it because for $\omega > 55$ $\Lambda$ is not defined?

- Lines 262-263 / Fig 13: I don't think that using the cumulative divide length is the most effective way to determine what are the orders changing the most (why not to use length($\omega$)) since by construction the curve shown in Fig 3 will be robust to changes for values corresponding to small $\omega$ in comparison with large $\omega$. I would be also interesting to see a similar curve when the analysis is done using the Strahler scheme, which I believe is more robust in terms of assigned orders (particularly to high order segments), to verify that indeed high order segments are the most affected. (Also lines 364-365)

- Lines285-290: There are some statements about dependencies between variables. Are those just inferred from visual inspection?

- Line 314-315: resembling instead of mimicking?

- Lines 332-333: *"Furthermore, the junction connectivity along the eastern edge is relatively low, despite being high up in the divide network (Figure 20a)."* Very qualitative statement and not that obvious from visual inspection, could you further quantify this point?

- Line 355: topology instead of geometry?

- Figure 4. The green and red tones are not that easy to distinguish (at least in a printed copy).

- Figure 18: I wonder whether the higher variability observed for the first 5 km is partially due to the higher density of points in that range. Does the same observation hold when distance percentiles are used on the x-axis (instead of the linear axis)?

- Fig 20: The layout/background used in panel a makes it difficult to delineate the actual catchment and compare with Fig 19a for example. enditemize

---

## Author Comment (AC1) · 21 Feb 2020

Author responses to the comments of the reviewers.

We first of all thank the reviewers for their effort and critical evaluation of our manuscript. We appreciated the constructive nature of the comments and tried our best to address them in the revised version of the manuscript.

Both reviewers found that the overall presentation of our study could be improved by making it shorter and focusing on specific aspects of the study. Reviewer 1 suggested to focus on the algorithm and metrics and presenting the numerical experiments in a separate paper. We think that is a great suggestion and thus we divided our study into two separate manuscripts. In part 1 we present the new algorithm and metrics, and apply it to the Big Tujunga to demonstrate capabilities and limits. In part 2 we present the landscape evolution experiments and our analysis of the response of the drainage divide network to perturbations. We think this separation helps focusing on the two aspects of the study. Although the two manuscripts together are even longer than the original submission, the individual manuscripts now have standard lengths and we think that the thread of each is much easier to follow.

In the following we provide all of the referee's comments, followed by our response where applicable. Because the reviewer's comments are related to the original submission, which is now presented in two separate papers, we provide our responses in blue font when related to part 1 of the revised version, and in red font, when related to part 2 of the revised version.

**Reviewer #1**

Referee report on "Identification and ordering of drainage divides in digital elevation models" by Scherler & Schwanghart

The manuscript submitted by Scherler & Schwanghart proposes a new way to calculate and characterize drainage basin boundaries and shows how the new metrics can be used to identify sections of the divide, which are vulnerable to topographic changes.

The paper presents a valuable contribution to geomorphology especially as the tools are made freely available in a easily usable framework (topotoolbox), which is accessible also for IT-skilled geomorphologists.

While the technical part of the manuscript is well presented and the algorithms clearly described, the analysis section lacks a central theme. Instead of getting to a point which shows how landscape perturbations affect the geometry or better topology of drainage divides the authors rush in a rather descriptive way through a series figures (Fig 11-21) most with multiple sub-pannels.

Given that the title focuses on "algorithm and metrics" with the analysis only being an "application showcase", I do not see this as a critical part for the paper. However the authors may consider shortening this part and focusing on only one perturbation scheme (maybe the rotating one which shows the most striking responses) which exemplifies their metrics without trying to make a geomorphological claim.

A more thorough analysis of different perturbations and their impact on drainage divide networks may be presented in a separate paper.

We thank the reviewer for her/his critical assessment of our study and the suggestions how to improve its presentation. We like the idea of focusing on the algorithm and metrics part and moving the impact of perturbations to a separate paper very much and therefore adopted it as outlined above.

Below a few minor comments and questions/suggestions:

For the efficient calculation of watersheds, as well as their susceptibility to perturbations by topographic changes, I would like to draw the author's attention to a series of papers which may not be on the general "radar" of the geomorphology / hydrology audience:

Fehr, E., et al. "New efficient methods for calculating watersheds." Journal of Statistical Mechanics: Theory and Experiment 2009.09 (2009): P09007.

Impact of Perturbations on Watersheds E. Fehr, D. Kadau, J. S. Andrade, Jr., and H. J. Herrmann Phys. Rev. Lett. 106, 048501

Scaling relations for watersheds E. Fehr, D. Kadau, N. A. M. Araujo, J. S. Andrade, Jr., and H. J. Herrmann Phys. Rev. E 84, 036116

We thank the reviewer for pointing out these interesting articles. They were indeed not on our radar. We added two citations to the manuscript.

Usually, when making a topographic analysis stream networks are defined by a flow accumulation threshold. The authors define a parameter which describes the distance from the divide to the closest stream. Such a parameter would, however, depend on the specific choice of the flow accumulation threshold and could vary between 0 and the system size. The authors should add a formal definition of their stream networks and clarify how they addressed this point.

Although we stated the flow accumulation threshold in the beginning of section 4.1, we missed to do so in the case of the numerical experiments. To avoid any confusion, we now provide the flow accumulation threshold (0.1 km$^2$ for the natural example and 0.2 km$^2$ for the numerical experiments) that we used to define the stream network in the method sections 3.2 and 2.2.

The following comments/questions are using Figures as reference, but should also be addressed in the corresponding parts of the main manuscript.

Figure 1: Wouldn't it make sense to put (a) and (b) into a single (bigger) figure such that the reader directly can associate the divide network with the corresponding stream network?

Yes, makes sense. Changed as suggested.

Figure 3: Would it be possible to reduce the complexity of the figure without scarifying the main message. If I read the figure correctly, the right hand half contains already all necessary information.

Yes, possible. Changed as suggested.

Fig 7: The 3d figures are small hard to understand.

a. Initialize: I would suggest the authors to put an arrow "time" from the initial to the final (steady state condition). (this may also be presented simply in plain top view) c. Labeling the (rotating) circle and 12h initial position and showing how this marker moves through time (3 snapshots) may be easier to understand than the small 3d visualization d. and e. I had first
difficulties to understand these graphics as they show the LEM surface and the "lithology" underneath, which changes the erodibility as the landscape erodes through this heterogeneous bottom. Here the authors may improve they description explicitly mentioning (depth) and the fact that the topography erodes through these formation creating a variable erodibility over time.

another option would be to combine c. d. e. as small panels in Fig. 11 and showing there 2 time snapshots in the evolution of the topography.

These are all great suggestions. Thanks! To improve the readability of this figure, we added a downward pointing arrow that indicates time as suggested and explained in the figure caption in more detail what is shown: "*The starting condition of each model is shown at the top of each rock column. During the experiments, the entire rock column is eroded and deeper lying regions are exhumed.*"

Fig 13: While the "Initialize" plot shows a significant change over the course of 10Myr, the changes in the plots of the perturbed systems seems almost insignificant, especially when compared to the changes of the "Reference" topography offer the course 10Myr-20Myr. Note, if I am correct, the "Reference" solution should have reached steady state and therefore should show no changes at all. In order to present how the disturbance changes affect the divides it would make more sense to compare the disturbed system to the reference system at the same time, thus t=15Myr and 20Myr and show the difference
between the two. Alternatively one could also evaluate the time evolution of the different perturbed cases relative to the evolution of the "Reference" solution.

We agree with the reviewer that the changes in the perturbed models are small compared to the initialize model. In the reference model, relatively small changes were still ongoing for quite some time, although standard topographic metrics, such as mean elevation, mean slope, etc. have reached steady state. We realized that referring to a simulation as reference
when there were still changes going on is confusing and we thus decided to run our models again. This time however, we extended the initialize simulation to a duration of 30 Myr, after which no more changes in the topography or drainage divide network occurred. The reference simulation is thus free of any divide migration.

When comparing the perturbed models with the reference model, we still prefer to show the entire time series. When focusing on one particular year, then only the particular difference between the drainage divide networks during that year
becomes visible, but not the range of differences observed during the models. In fact, it's really the temporal variation in the network that makes the difference and that requires showing not just one year.

Note also that we changed the y-axis scale from cumulative divide length to divide length, in response to a comment from Reviewer 2 regarding this figure. This change improves the conspicuity of the differences between the reference and the perturbed models.

Fig 17: Fig 17b consists of a main (invariant) cluster (black dashed line) corresponding to the general topographic shape of the landscape. It would be interested to see where (in the landscape) the deviations from this behavior are located.

That's a great suggestion. We tried to incorporate it by marking those junctions that correspond to the points lying outside of the invariant cluster in a new panel. We tested adding such markers to the existing panels, but quickly realized that the figure gets overcrowded this way. We therefore decided to add another panel in which we show the junctions, their elevation, and
the markers that indicate which junctions lie outside the range of values observed in the Reference model.

Fig 21: The scatter plots in this figure are difficult to interpret, especially because color is an additional variable. e.g. do yellow points mask blue markers underneath? It may be favorable to divide the color scale in 3 or 4 categories and bin the results in x,y for each category. Additionally the authors should quantify between the different measured quantities using e.g. a simple linear regression model.

Yes, we can see the point. It is true that yellow points mask blue points underneath and we have been discussing amongst us for quite some time how to best present the pattern. Categories would only help when combining them with an additional layer of information, like bins, as suggested, but that would further increase the complexity of the figure, especially if added to each axis of each of the panels. To help the reader understand the figure, we instead added a note to the caption, where we point out that the data points have been sorted to show the highest divide distance above data points with lower divide distance, and that data points with lower divide distance occur beneath those with higher divide distance. We hope that this explanation avoids confusion or the impression we would try to obscure something.

   We furthermore added a new figure to the revised version, in which we quantified the correlation between the different metrics and erosion rate differences using a linear regression model, as suggested by the reviewer. In this figure, we show how the coefficient of determination ($R^2$ value), differs between the models, the metrics and also depends on the divide distance. We think this is a very useful figure that avoids some of the limitations we were facing with the other figure that shows merely the scatter plots.

   I hope that the above comments & suggestions help the author to improve the manuscript and strengthen their arguments.

   They surely did – thank you very much!

**Reviewer #2**

   The authors present an algorithm to automatically extract the drainage divide networks from digital elevation models as well as ordering schemes to classify the different divides with respect to each other. The application of the algorithm to a field example (San Gabriel Mountains) and different numerical landscapes is shown, exploring the relationship between different geometric and topographic metrics attributed to divides with respect to their mobility.

   I believe that the tool presented in this manuscript (and already implanted in Topotoolbox) is valuable to explore landscapes more systematically, offering the potential of developing new approaches to characterize landscape stability and to retrieve information about the history and drivers of landscape evolution.

   I also believe that Esurf is the perfect outlet for this contribution, but before I can recommend the manuscript for publication,

I would like to point out some comments/concerns for the authors to consider in order to clarify certain parts of the manuscript, as well as to highlight the main points of this contribution.

   We thank the reviewer for her/his assessment of our study and the suggestions how to improve its presentation.

**1 General Comment**

   The authors have carried out an extensive analysis of divide drainage networks from a field case (Big Tujunga catchment)

and several numerically simulated landscapes (including their temporal evolution) by exploring different (topologic, geometric and topographic) metrics. The manuscript contains 21 figures (with 120+ panels! – some of them quite complex). I think the number of figures is excessive, particularly because the presence of many of them (and their description) are not that insightful/necessary to make the main points on the paper, which are currently diluted. I would strongly suggest that the authors transfer to the Supplementary Material most of the panels, keeping only the ones that are key to the message, and
expanding in their description and insight gained from this analysis. I believe the manuscript will substantially benefit from
an effort in this direction, improving its readability and impact.

We thank the reviewer for the open criticism. The main message we take out from the above paragraph is similar to what
reviewer one also commented. In its present form the paper does not manage to convey the main points because it branches
into too many directions. Instead of moving figure panels to the supplementary material, we better like the idea of splitting
the paper into two parts; one that focuses on the algorithm and the metrics, and one that analyses the response of the drainage
network to perturbations. See our initial statements at the beginning of this document. For the paper that focuses on the
numerical experiments (part 2), we prefer to retain the different models and present their results next to each other in
different panels. We find that this way of presenting the results facilitates comparing the results of the different numerical
simulations.

**2 Comments:**

- Line 76: It is stated that divide networks even contain cycles. Do cycles correspond to enclosed (internally drained)
  basins? In the rest of the manuscript, it is said that these networks are represented as tree-like networks, i.e. no
  cycles, (e.g. line 116) and that enclosed basins are not considered in the analysis. Is the algorithm and ordering
  scheme applicable for the cases the network contains cycles?

Ouch! First question and already touched the most sensitive part of the algorithm! Yes, by cycles we referred to
internally drained basins, and no, they don't reflect a true tree network. Yes, we did not consider internally drained
basins in our analysis, and no, the ordering scheme is not applicable for that case.

But more formally: as stated in line 162 (page 6) of the original submission, our algorithm does not handle internally
drained basins well and in all of our examples they don't show up. The problem simply is that the sorting procedure fails
when the divide of an internally drained basin is encountered. Nevertheless, we are working on a solution that allows
bypassing internally drained basins in order to reach divides beyond them, where the condition is met again, but this
proves more challenging than anticipated.

To make the reader aware of this limitation, we modified section 2 and we expanded the final paragraph of section 3.1,
in which we describe the divide algorithm: "*As aforementioned, our divide algorithm currently does not handle
internally drained basins. Whereas the divides of internally drained basins are easy to identify, they are not easily
sorted in a meaningful manner. In fact, the distance to a common stream location ($\Lambda$) is undefined for a divide of an
internally drained basin. At the moment, the sorting procedure (Figure 4) stops at such divides, because the divide
segments cannot be assigned a direction. In consequence, also parts of the divide network that potentially lie beyond an
internally-drained basin, and for which the distance to a common stream location is defined, cannot be reached
anymore. While we are working on a solution to this issue, our algorithm is currently best applied to acyclic drainage
divide networks.*"

- A new ordering scheme is introduced (Topo) and it is mostly adopted for the analysis reported in the manuscript. However, I missed a more detailed comparison between the different schemes in terms of characterizing the divide network.

We have modified the entire text in several places, which also affected the presentation and discussion of the ordering schemes in sections 3.1, 4.1, and 5.1. We hope that this level of detail is sufficient.

- There are some metrics used or introduced in the manuscript, which although might seem intuitive, are not properly defined, and they can lead to misunderstandings. I would encourage the authors to properly define each of the metrics, particularly those that are central to the manuscript. As an example, I have struggled in imagining what is the definition of divide distance (which is used in almost every figure on the paper). The first time that it is mentioned in the manuscript is in line 129-131, where it is stated: *"We thus propose to order the nodes and edges of the divide network, by their maximum down-divide distance from divide endpoints, measured either in map units or in the number of divide segments. From now on, we call the distance measured in map units along the divide network the divide distance (dd)."* What does *down-divide distance* mean? Only paths where elevation is strictly decreasing are considered? Please provide a clear definition of the different metrics utilized (consider including an illustration if needed – e.g. Fig 4 is quite helpful)

That's a very good point! We agree that down-divide distance is a term not easily understood. It does not refer to elevation, but is related to assigning directions to the divide segments and measuring distance along these directions. We modified section 2 in several places to provide a clear definition of all terms that we introduce. Specifically, we explain in much more detail how we can go from a drainage divide (section 2.1) to a divide network (section 2.2). First, we point out that divides have no terrain property that allows to easily assign directions to the divide network: *"In contrast, drainage divides have no inherent direction, and there exists no terrain property, like elevation, that could be used to assign a direction to them."* We then explain how directions can be assigned based on the tree-like network topology: *"Instead, we suggest that directions can be derived from the tree-like structure of drainage divide networks. Analogous to a parcel of water that travels down a river from its source to its mouth, we propose to start at the leaves of the tree, which we call the endpoints of the divide network (Figure 2), and incrementally move down the branches (Figure 4). Note that the term "move down" does not refer to elevation, but to the hierarchy of the divide network. Where two or more drainage divides meet, they form a junction. We call individual parts of drainage divides that link endpoints and junctions, junctions and junctions, or endpoints and endpoints, the drainage-divide segments, and we refer to the ends of divide segments as segment termini to avoid confusion with endpoints. At junctions with more than one unsorted divide segment, the sorting process pauses because it is not obvious in which direction the sorting shall continue. However, in the absence of cycles (internally drained basins), each junction will reach a point in the sorting loop when there is only one unsorted divide segment left so that the sorting can continue (Figure 4). This condition assures that the divide segments are correctly sorted in a tree-like manner, but it fails when encountering a cycle. As we will show later, the average branch length $\Lambda$ scales linearly with the maximum distance from an endpoint along the sorted divide network, as well as the maximum number of divide segments (or junctions), both of which are more easily computed. We thus propose to order the nodes and edges of the divide network, by their maximum distance from divide endpoints, measured either in map units or in the number of divide segments. From now on, we call the distance measured in map units along the directed divide network the divide distance (dd)."* To illustrate our points, we call out to the former Figure 5, which now appears before the former Figure 3.

- Section 3, particularly section 3.1 and Appendix 7.1, are the central contribution in the paper. However, I find the description of the algorithm not very clear, particularly the second step (lines 140-143) and Figure 3, which I am unable to follow. There are terms that are not defined, e.g. how do you define (the number of) segment termini?

Agreed. We added a definition of segment terminus to section 2 and labelled a divide segment in Figure 2. We also modified section 3.1 and explained our algorithm in more detail. We support this explanation with a figure that shows the workflow of our approach and hope that the different steps of the algorithm are now easier to understand.

- Line 181 – Since the metrics are computed for each divide edge, isn't $\square$X just twice the value of $\bar{X}$? In that case, I guess there is no need to keep both.

Yes, it's true that the sum of a metric is twice the mean of that metric, but we don't think there is need to get rid of either the sum sign and replace it with two times the mean or to replace the mean sign with the sum divided by two.

- Equation 7. Not clear. Should d and $d_d$ function of x within the integral?

Thanks for spotting this error. The equation was actually incorrect. We checked our procedure and replaced it with the following equation: $C_J = \frac{dx}{d_{d,max}} \sum_{i=1}^{n} \frac{d_i}{d_{d,i}}$; and also changed the accompanying text: "*We define $C_J$ to correspond to the sum of the ratios of the Euclidean distance, d, and the divide distance, $d_d$, integrated over of all divide edges, n, within a specified maximum divide distance, $d_{d,max}$, times the ratio of the cell size, dx, and $d_{d,max}$.*"

- Line 240-241:"*In contrast, the ordering rules of the Strahler and Shreve scheme (see Eq. 1 and Eq. 2) may yield unequal orders during the sorting, so that the divide orders of the last divide segments may be different by more than one.*" I can see how this can happen for the Shreve scheme but under what circumstances would that occur using the Strahler scheme?

Under the same circumstance. The Strahler scheme requires two joining segments of equal order for the continuing segment to increase in order. If only one has a higher order, the order does not increase. In the Big Tujunga catchment, the part north of the main stem river has greater area and thus larger tributaries and longer divides that climb up to higher orders compared to the southern part. The Strahler orders along the northern drainage divide of the entire Big Tujunga catchment thus increase more rapidly to higher Strahler orders and eventually reach a higher Strahler order than they do along the southern divide (see figure below). This explanation is the same that we gave in the text for why the 250 Shreve order increases more rapidly and we thus modified the text to make it clear that this explanation applies for the Strahler scheme, too.

[Figure]

Figure caption: Drainage divide network of the Big Tujunga catchment. Divide orders derived from Strahler ordering scheme: Green = 1, purple = 2, yellow = 3, red = 4, blue =5.

- Line254–Why the analysis is limited to $\omega<55$. Is it because for $\omega>55$ $\Lambda$ is not defined?

Yes, exactly. The extent of the DEM is not large enough for defining $\Lambda$ at $\omega>55$. As stated in the text (section 2) that would be one of the weaknesses of using $\Lambda$, and the reason why we refer to the divide distance instead. We added an explanation to the text: "*Although the maximum order for which $\Lambda$ can be determined (because of the size of our DEM) is limited to $\omega \leq 55$...*"

- Lines 262-263 / Fig 13: I don't think that using the cumulative divide length is the most effective way to determine what are the orders changing the most (why not to use length($\omega$)) since by construction the curve shown in Fig 3 will be robust to changes for values corresponding to small $\omega$ in comparison with large $\omega$. I would be also interesting to see a similar curve when the analysis is done using the Strahler scheme, which I believe is more robust in terms of assigned orders (particularly to high order segments), to verify that indeed high order segments
are the most affected. (Also lines 364-365)

Great point. We changed the y-axis, which now shows the divide length for a given divide order. When using the Strahler ordering scheme, there are only very few different divide orders. Changes in the network structure are thus more difficult to observe, because they could occur within one divide order, whereas they are represented by different divide orders in the Topo ordering scheme. In any case, we think that the new models, specifically the Reference model,
which shows no more changes, and the way we now present the divide length by order, provide good support for the observation that mostly the distributions of high order segments are affected by perturbations.

- Lines285-290: There are some statements about dependencies between variables. Are those just inferred from visual inspection?

Yes, they are inferred from visual inspection of Figure 15 (now Figure 8 in the revised version). We think that statistical tests to determine whether the values observed for different divide distances are truly distinct is not required for the argument. We note however that the dependencies of some metrics on divide distance that we describe are much clearer in the new models, specifically the Reference model, in which nothing happens, and where the sentences are related to. The dependencies of the other metrics (cross-divide differences in topographic metrics) are further studied in figures 11 and 12.

- Line 314-315: resembling instead of mimicking?

Changed to "resembling".

- Lines 332-333: *"Furthermore, the junction connectivity along the eastern edge is relatively low, despite being high up in the divide network (Figure 20a)."* Very qualitative statement and not that obvious from visual inspection, could you further quantify this point?

Good point. We deleted this statement.

- Line 355: topology instead of geometry?

Changed to "topology".

- Figure 4. The green and red tones are not that easy to distinguish (at least in a printed copy).

We increased the size of the data points. Hopefully that helps.

• Figure 18: I wonder whether the higher variability observed for the first 5 km is partially due to the higher density of points in that range. Does the same observation hold when distance percentiles are used on the x-axis (instead of the linear axis)?

That's a good point. We added a statement that acknowledges the larger number of observations for lower divide
distances: "*It should be noted that divide edges at low divide distances are much more abundant compared to those at higher distances, or equivalently, at higher divide orders (Figure 7)*". Regarding the x-axis scaling: Each divide node has a divide distance that increases when moving up in the divide network structure. Because the divide distance is defined to be the maximum distance from an endpoint along the directed divide network, there exist jumps in divide distance. When referring to the entire divide network for computing distance percentile, the picture does not change, only the labeling of the x-axis. When referring to individual divide segments, for example, the whole figure gets very
messy, as both short and long divide segments, anywhere within the network will plot on top of each other. However, the greater scatter for lower divide orders/distances is a stable observation that is also a common observation in the numerical experiments in part 2.

- Fig 20: The layout/background used in panel a makes it difficult to delineate the actual catchment and compare with Fig 19a for example.

We removed this figure in the revise version.

---

## Author Response (AR2)

Dear editors,

Thank you for the positive evaluation of our manuscript. We applied almost all of the corrections listed below. In some cases we also provided the sentence that we added or changed, if it was not just a comma issue or something like that, or a response if no changes were needed – in our opinion.
If there are still remaining issues, we can hopefully handle it in the proofs stage.

Thank you for your help!

The two authors.

Dear authors,

I am pleased to confirm acceptance of your work for publication in esurf, now in the form of a 2-part series of papers. I appreciate you careful and thoughtful attention to the comments from the reviewers and AE. Like the AE, I think splitting the manuscript into two parts was a good outcome of this process.

As noted by the AEs and reviewers, this work adds a great deal to our toolkit for analyzing topography, and to our general understanding of divides and divide migration -- a topic of great interest in recent literature. Thank you for choosing esurf as venue (and I am sorry the final decision has been slightly delayed, while I was clarifying how it would work to have your single original submission published as two separate papers -- that should be fine from here).

I have a number of very small editing suggestions below, including a few grammatical corrections. You might consider making these corrections prior to submitting your final files for typesetting; alternatively, most could be addressed during copy editing if you do not want to delay further.

Thank you again!

Part 1

Line 17: no comma after junctions
Line 20: "a stream and hillslope relief," (remove the and add comma)
Line 22: in addition to divide instability, could such differences relate to structural controls (e.g., for tilted sedimentary layers)? if so, perhaps worth qualifying slightly here
RESPONSE: no, the rocks in this area of the San Gabriel Mountains are dominated by crystalline rocks without strong foliation or bedding that could induce such asymmetry.
Line 37: I think "eastern Tibetan Plateau" (small e)
Line 55: comma after directions
Line 92: no comma after one
Line 108-109: I think it would be useful to add a parenthetical comment after this sentence to define what you mean by "cycles" (I assume that this refers to divides that form complete closed loop, or something similar, but a formal definition would help readers, I think)
RESPONSE: hmm, the manuscript version we submitted actually contains exactly that: "*Analogous to streams, drainage divides are typically organized into tree-like networks (Figure 1), although cycles that correspond to internally-drained basins may exist.*"
Line 131: no comma after network
Line 140: odd wording; perhaps better as "the more detailed the stream and divide networks will be"
Line 168: comma not semicolon after segments
Line 170: no comma after Rb
Line 184:
Line 186: comma not semicolon after the citation
Line 208: "As a consequence"
Line 228: "increases linearly as 0.36 x divide order" (as not with seems more appropriate)
Line 238: check for missing superscript 2 on 1 km^2
RESPONSE: is there a conversion issue? In the manuscript we submitted (pdf), the superscript exists.

Line 239: I don't think this is a good use of cf. (generally this abbreviations means "compare to"; you have used it in contexts that seem like you are asking readers to refer to a reference or figure, not compare to it, so I am encouraging you to rethink its use)
Lines 246-247: "averages… are" or "average… is" (perhaps the former)
Line 258: see note about regarding use of cf.
Line 261: "enquire" doesn't seem right here; perhaps "query"?
Line 277: see note about regarding use of cf.
Line 278: "its", not "it's"
Line 288: you say "or" here, but I would have thought this could be "and"?
Line 299: no comma after scheme
Line 335: "lends itself to topological analysis"
Figure 1 caption: specify which type of divide order is shown here? (Topo, I think?)
Figure 1: consider adding a north arrow?
Figure 3 caption: specify what the black lines are; also maybe note where this topography is from?
Figure 4 caption: no comma after remaining
Figure 9 caption: no comma after order
Figure 10 caption: check for missing superscript 2 on 1 km^2
RESPONSE: is there a conversion issue? In the manuscript we submitted (pdf), the superscript exists.
Figure 12: maybe consider labeling Chilao Flats on the map and/or images, since it receives particular attention in the Discussion?

Part 2

Lines 13-14: I found this confusing; you might consider rewording, perhaps along the lines of, "when considering divide segments with divide end points that are a minimum distance of ~5km from river confluences." Or something like that…
RESPONSE: sentence changed to: *"Results show that divide segments that are a minimum distance of ~5 km from river confluences strive to attain constant values of hillslope relief as well as flow distance to the nearest stream."*
Line 31: argued, not argue (to keep tense the same in this paragraph)
Line 34: argued, not argue (as above)
Line 39: holds, not hold (shape is singular)
Line 44: no comma after mobility
Line 49: I wonder if you could add a short sentence here to point out the features of your new analysis tool that allows you to do more than has been done before? You allude to this later in the paragraph, but could perhaps be explicit about how what you is enabled by your new capabilities.
RESPONSE: yes, good idea. We added the following: "*…, which removes the necessity to manually select drainage divides for comparison.*"
Line 60: and, not as well as
Line 75: see note about regarding use of cf.
Line 79: I am unclear about the meaning of the "-" in "50-%"; is this supposed to refer to the fact that you reduced the quantity by 50%? In this case I think it is clear just to say "50% reduced"
Line 80: "representative of"
Line 81: no comma after landscape
Line 83: "resulted in the resistant layers regularly sweeping from"
Line 84: see note above about "-" in "75-%"
Line 92: see note about regarding use of cf.
Line 111: see note about regarding use of cf.
Line 116: no comma after distance
Line 153: I find the meaning of "factor on divide order" slightly unclear
RESPONSE: changed to "The 430 m corresponds to the mean length of the divide segments."
Line 157: no comma after parameters
Line 165: are, not is
Line 177: this wording is awkward; perhaps "attains temporally-averaged values in the perturbed models that are quite similar to the constant values"
Line 192: add a comma after notable
Lines 209-201: "occupy the centre [OF WHAT?] as well as the centers of the four quadrants" (specify of what)
Line 212: see note about regarding use of cf.
Line 224: "recording similar" is a bit unclear; consider rewording?

RESPONSE: changed to "recording similar tectonic forcing".
Line 228: see note about regarding use of cf.
Line 232: "has become a popular tool" may be a bit better wording?
Line 234: remove "eventually"
Line 236: "and", not "as well as"
Line 246: see note about regarding use of cf.
Line 248: comma after bins
Line 255: perhaps "these values and their variability"?
Line 259: demands, not demand (scatter is singular)
Lines 266-267: it struck me that stochastic divide migration (e.g., mass wasting events) adds another complexity in natural systems that is not captured in this version of the numerical model (I think?)
RESPONSE: yes, good point, we included mass wasting events.
Line 275: "and long divide distances"?
Line 279: comma, not semicolon
Line 192: no comma after (Cj)
Line 293: consistent, not consistence
Line 303: remove "thus"?
Figure 1: add north arrow, and specify UTM zone in caption?
RESPONSE: we added the north direction and UTM zone to the caption.
Figure 2 caption: see note above regarding "50-%" and "75-%"